# Meta-Learning Adversarial Bandit Algorithms

**Mikhail Khodak**[*]
Carnegie Mellon University
khodak@cmu.edu

**Ilya Osadchiy**[*]
Technion - Israel Institute of Technology
osadchiy.ilya@gmail.com

**Keegan Harris**
CMU

**Maria-Florina Balcan**
CMU

**Kfir Y. Levy**
Technion

**Ron Meir**
Technion

**Zhiwei Steven Wu**
CMU

## Abstract

We study online meta-learning with bandit feedback, with the goal of improving performance across multiple tasks if they are similar according to some natural similarity measure. As the first to target the adversarial online-within-online partial-information setting, we design meta-algorithms that combine outer learners to simultaneously tune the initialization and other hyperparameters of an inner learner for two important cases: multi-armed bandits (MAB) and bandit linear optimization (BLO). For MAB, the meta-learners initialize and set hyperparameters of the Tsallis-entropy generalization of Exp3, with the task-averaged regret improving if the entropy of the optima-in-hindsight is small. For BLO, we learn to initialize and tune online mirror descent (OMD) with self-concordant barrier regularizers, showing that task-averaged regret varies directly with an action space-dependent measure they induce. Our guarantees rely on proving that unregularized follow-the-leader combined with two levels of low-dimensional hyperparameter tuning is enough to learn a sequence of affine functions of non-Lipschitz and sometimes non-convex Bregman divergences bounding the regret of OMD.

## 1 Introduction

Learning-to-learn [51] is an important area of research that studies how to improve the performance of a learning algorithm by *meta-learning* its parameters—e.g. initializations, step-sizes, and/or representations—across many similar tasks. The goal is to encode information from previous tasks in order to achieve better performance on future ones. Meta-learning has seen a great deal of experimental work [24, 49], practical impact [21, 29], and theoretical effort [11, 18, 22, 45, 20]. One important setting is online-within-online meta-learning [19, 31], where the learner performs a sequence of tasks, each of which has a sequence of rounds. Past work has studied the *full-information* setting, where the loss for every arm is revealed after each round. This assumption is not realistic in many applications, e.g. recommender systems and experimental design, where often partial or *bandit* feedback—only the loss of the action taken—is revealed. Such feedback can be *stochastic*, e.g. the losses are i.i.d. from some distribution, or *adversarial*, i.e. chosen by an adversary. We establish the first formal guarantees for online-within-online meta-learning with adversarial bandit feedback.

As with past full-information meta-learning results, our goal when faced with a sequence of bandit tasks will be to achieve low regret *on average* across them. Specifically, our task-averaged regret should (a) be no worse than that of algorithms for the single-task setting, e.g. if the tasks are not very similar, and should (b) be much better on tasks that are closely related, e.g. if the same small set of arms do well on all of them. We show that a natural way to achieve both is to initialize and tune online mirror descent (OMD), an algorithm associated with a strictly convex regularizer whose hyperparam-

---

[*] Denotes equal contribution.

37th Conference on Neural Information Processing Systems (NeurIPS 2023).

eters have a significant impact on performance. Our approach works because it can learn the best hyperparameters in hindsight across tasks, which will recover OMD's worst-case optimal performance if the tasks are dissimilar but will take advantage of more optimistic settings if they are related. As generalized distances, the regularizers also induce interpretable measures of similarity between tasks.

## 1.1 Main contributions

We design a meta-algorithm (Algorithm 1) for learning variants of OMD—specifically those with entropic or self-concordant regularizers—that are used for adversarial bandits. This meta-algorithm combines three *full-information* algorithms—follow-the-leader (FTL), exponentially weighted online optimization (EWOO), and multiplicative weights (MW)—to set the initialization, step-size, and regularizer-specific parameters, respectively. It works by optimizing a sequence of functions that each *upper-bound* the regret of OMD on a single task (Theorem 2.1), resulting in (a) interesting notions of task-similarity because these functions depend on generalized notions of distances (Bregman divergences) and (b) adaptivity, i.e not needing to know how similar the tasks are beforehand.

Our first application is to OMD with the Tsallis regularizer [3], a relative of Exp3 [6] that is optimal for adversarial MAB. We bound the task-averaged regret by the Tsallis entropy of the *estimated* optima-in-hindsight (Corollary 3.1), which we further extend to that of the *true* optima by assuming a gap between the best and second-best arms (Corollary 3.2). Both results are the first known consequences of the online learnability of Bregman divergences that are *non-convex* in their second arguments [31], while the latter is obtained by showing that the loss estimators of a modified algorithm identify the optimal arm w.h.p. As an example, our average $m$-round regret across $T$ tasks under the gap assumption is

$$o_T(\text{poly}(m)) + 2 \min_{\beta \in (0,1]} \sqrt{H_\beta d^\beta m / \beta} + o(\sqrt{m}) \tag{1}$$

where $d$ is the number of actions and $H_\beta$ is the Tsallis entropy [52, 3] of the distribution of the optimal actions ($\beta = 1$ recovers the Shannon entropy).[1] This entropy is low if all tasks are usually solved by the same few arms, making it a natural task-similarity notion. For example, if only $s \ll d$ of the arms are ever optimal then $H_\beta = \mathcal{O}(s)$, so using $\beta = 1/\log d$ in (1) yields an asymptotic task-averaged regret of $\mathcal{O}(\sqrt{sm \log d})$, dropping fast terms. For $s = \mathcal{O}_d(1)$ this beats the minimax optimal rate of $\Theta(\sqrt{dm})$ [5]. On the other hand, since $H_{1/2} = \mathcal{O}(\sqrt{d})$, the same bound recovers this rate in the worst-case of dissimilar tasks.

Lastly, we adapt our meta-algorithm to the adversarial BLO problem by setting the regularizer to be a self-concordant barrier function, as in Abernethy et al. [2]. Our bounds yield notions of task-similarity that depend on the constraints of the action space, e.g. over the sphere the measure is the closeness of the average of the estimated optima to the sphere's surface (Corollary 4.1). We also instantiate BLO on the bandit shortest-path problem (Corollary 4.2) [50, 30].

## 1.2 Related work

While we are the first to consider meta-learning under adversarial bandit feedback, many have studied meta-learning in various *stochastic* bandit settings [9, 34, 47, 48, 35, 13, 15, 41, 10]. The latter three study stochastic bandits under various task-generation assumptions, e.g. Azizi et al. [10] is in a batch-within-online setting where the optimal arms are adversarial. In contrast, we make no distributional assumptions either within or without. Apart from this difference, the results of Azizi et al. [10] are the ones our MAB results are most easily compared to, which we do in detail in Section 3. Notably, they assume that only $s \ll d$ of the $d$ arms are ever optimal across $T$ tasks and show (roughly speaking) $\tilde{\mathcal{O}}(\sqrt{sm})$ asymptotic regret; we instead focus on an entropic notion of task-similarity that achieves the same asymptotic regret when specialized to their $s \ll d$. However, avoiding their explicit assumption has certain advantages, e.g. robustness in the presence of $o(T)$ outlier tasks (c.f. Section 3.3).

A setting that bears some similarity to online-within-online bandits is that of switching bandits [6], and more generally online learning with dynamic comparators [4, 27, 38, 7, 55]. In such problems, instead of using a static best arm as the comparator we use a piecewise constant sequence of arms, with a limited number of arm switches. The key difference between such work and ours is our assumption that task-boundaries are known; this makes the other setting more general. However, while e.g. Exp3.S [6] can indeed be applied to online meta-learning, guarantees derived

---

[1]We use $\mathcal{O}_n(\cdot)$ (and $o_n(\cdot)$) to denote terms with constant (and sub-constant) dependence on $n$.

from switching costs *cannot* improve upon just running Tsallis-INF on each task [39, Table 1]. Furthermore, these approaches usually quantify difficulty by the number of switches, whereas we focus on task-similarity. While there exists stochastic-setting work that measures difficulty using a notion of average change in distribution across rounds [53], it does not lead to improved performance if this average change is $\Omega(T)$, as is the case in e.g. the $s$-sparse setting discussed above.

There has been a variety of work on full-information online-within-online meta-learning [32, 12], including tuning OMD [31, 19]. Doing so for bandit algorithms has many additional challenges, including (1) their inherent and high-variance stochasticity, (2) the use of non-Lipschitz and even unbounded regularizers, and (3) the lack of access to task-optima in order to adapt to deterministic, algorithm-independent task-similarity measures. Theoretically our analysis draws on the average regret-upper-bound analysis (ARUBA) framework [31], which observes that OMD can be tuned by targeting its upper bounds, which are affine functions of Bregman divergences, and provide online learning tools for doing so. Our core structural result shows that the distance generating functions $\psi_\theta$ of these Bregman divergences can be tuned without interfering with meta-learning the initialization and step-size; tuning $\theta$ is critical for adapting to settings such as that of a small set of optimal arms in MAB. Doing so depends on several refinements of the original approach, including bounding the task-averaged-regret via the spectral norm of $\nabla^2 \psi_\theta$ and expressing the loss of the meta-comparator using only $\psi_\theta$, rather than via its Bregman divergence as in prior work. Finally, applying our structural result requires setting-specific analysis, e.g. to show regularity w.r.t. $\theta$ or to obtain MAB guarantees in terms of the entropy of the true optimal arms. The latter is especially difficult, as Khodak et al. [31] define task-similarity via full information upper bounds, and involves applying tools from the best-arm-identification literature [1] to show that a constrained variant of Exp3 finds the optimal arm w.h.p.

## 2 Learning the regularizers of bandit algorithms

We consider the problem of meta-learning over bandit tasks $t = 1, \ldots, T$ over some fixed set $\mathcal{K} \subset \mathbb{R}^d$, a (possibly improper) subset of which is the action space $\mathcal{A}$. On each round $i = 1, \ldots, m$ of task $t$ we play action $\mathbf{x}_{t,i} \in \mathcal{A}$ and receive feedback $\ell_{t,i}(\mathbf{x}_{t,i})$ for some function $\ell_{t,i} : A \mapsto [-1, 1]$. Note that all functions we consider will be linear and so we will also write $\ell_{t,i}(\mathbf{x}) = \langle \ell_{t,i}, \mathbf{x} \rangle$. Additionally, we assume the adversary is *oblivious within-task*, i.e. it chooses losses $\ell_{t,1}, \ldots, \ell_{t,m}$ at time $t$. We will also denote $\mathbf{x}(a)$ to be the $a$-th element of the vector $\mathbf{x} \in \mathbb{R}^d$, $\mathcal{K}^\circ$ to be the interior of $\mathcal{K}$, $\partial \mathcal{K}$ its boundary, and $\triangle$ to be the simplex on $d$ elements. Finally, note that all proofs can be found in the Appendix.

In online learning, the goal on a single task $t$ is to play actions $\mathbf{x}_{t,1}, \ldots \mathbf{x}_{t,m}$ that minimize the regret $\sum_{i=1}^m \ell_{t,i}(\mathbf{x}_{t,i}) - \ell_{t,i}(\mathring{\mathbf{x}}_t)$, where $\mathring{\mathbf{x}}_t \in \arg\min_{\mathbf{x} \in \mathcal{K}} \sum_{i=1}^m \ell_{t,i}(\mathbf{x})$. Lifting this to the meta-learning setting, our goal as in past work [31, 19] will be to minimize the **task-averaged regret**: $\frac{1}{T} \sum_{t=1}^T \sum_{i=1}^m \ell_{t,i}(\mathbf{x}_{i,t}) - \ell_{t,i}(\mathring{\mathbf{x}}_t)$. In particular, we want to use multi-task data to improve average performance as the number of tasks $T \to \infty$. For example, we wish to attain a task-averaged regret bound of the form $o_T(\text{poly}(m)) + \tilde{\mathcal{O}}(V\sqrt{m}) + o(\sqrt{m})$, where $V \in \mathbb{R}_{\geq 0}$ is a measure of task-similarity that is small if the tasks are similar but still yields the worst-case single-task performance—$\mathcal{O}(\sqrt{dm})$ for MAB and $\mathcal{O}(d\sqrt{m})$ for BLO—if they are not.

### 2.1 Online mirror descent as a base-learner

In meta-learning we are commonly interested in learning a within-task algorithm or **base-learner**, a parameterized method that we run on each task $t$. A popular approach is to learn the initialization and other parameters of a gradient-based method such as gradient descent [24, 44, 36]. If the task optima are close, the best initialization should perform well after only a few steps on a new task. We take a similar approach applied to online mirror descent, a generalization of gradient descent to non-Euclidean geometries [14]. Given a strictly convex **regularizer** $\psi : \mathcal{K}^\circ \mapsto \mathbb{R}$, step-size $\eta > 0$, and initialization $\mathbf{x}_{t,1} \in \mathcal{K}^\circ$, OMD has the iteration

$$\mathbf{x}_{t,i+1} = \arg\min_{\mathbf{x} \in \mathcal{K}^\circ} B(\mathbf{x}||\mathbf{x}_{t,1}) + \eta \sum_{j \leq i} \langle \nabla \ell_{t,j}(\mathbf{x}_{t,j}), \mathbf{x} \rangle \tag{2}$$

where $B(\mathbf{x}||\mathbf{y}) = \psi(\mathbf{x}) - \psi(\mathbf{y}) - \langle \nabla \psi(\mathbf{y}), \mathbf{x} - \mathbf{y} \rangle$ is the **Bregman divergence** of $\psi$. OMD recovers online gradient descent when $\psi(\mathbf{x}) = \frac{1}{2}\|\mathbf{x}\|_2^2$, in which case $B(\mathbf{x}||\mathbf{y}) = \frac{1}{2}\|\mathbf{x} - \mathbf{y}\|_2^2$; another example is exponentiated gradient, for which $\psi(\mathbf{p}) = \langle \mathbf{p}, \log \mathbf{p} \rangle$ is the negative Shannon entropy on probability vectors $\mathbf{p} \in \triangle$ and $B$ is the KL-divergence [46]. An important property of $B$ is that the sum over functions $B(\mathbf{x}_t||\cdot)$ is minimized at the mean $\bar{\mathbf{x}}$ of the points $\mathbf{x}_1, \ldots, \mathbf{x}_T$.

**Algorithm 1:** Tunes $\text{OMD}_{\eta,\theta}$ with regularizer $\psi_\theta : \mathcal{K}^\circ \mapsto \mathbb{R}$ and step-size $\eta > 0$, which when run over loss estimators $\hat{\ell}_{t,1}, \ldots, \hat{\ell}_{t,m}$, yielding task-optima $\hat{\mathbf{x}}_t = \arg\min_{\mathbf{x} \in \mathcal{K}} \sum_{i=1}^m \langle \hat{\ell}_{t,i}, \mathbf{x} \rangle$.

---

**Input:** compact set $\mathcal{K} \subset \mathbb{R}^d$, initialization $\mathbf{x}_1 \in \mathcal{K}$, ordered subset $\Theta_k \subset \mathbb{R}$ also used to index
interval bounds $\underline{\eta}, \overline{\eta} \in \mathbb{R}^k_{\geq 0}$ and hyperparameters $\alpha \in \mathbb{R}^k_{\geq 0}$, scalar hyperparameters
$\rho > 0$ and $\lambda \geq 0$, learners $\text{OMD}_{\eta,\theta} : \mathcal{K} \mapsto \mathbb{R}^d$, projections $\mathbf{c}_\theta : \mathcal{K} \mapsto \mathcal{K}_\theta$

**for** $\theta \in \Theta_k$ **do**
    $\mathbf{w}_1(\theta) \leftarrow 1$ and $\eta_1(\theta) \leftarrow \frac{\underline{\eta}(\theta) + \overline{\eta}(\theta)}{2}$          `// initialize MW and EWOO`

**for** *task* $t = 1, \ldots, T$ **do**
    sample $\theta_t$ from $\Theta_k$ w.p. $\propto \exp(\mathbf{w}_t)$      `// sample from MW distribution`
    $\hat{\mathbf{x}}_t \leftarrow \text{OMD}_{\eta_t(\theta_t),\theta_t}(\mathbf{c}_{\theta_t}(\mathbf{x}_t))$        `// run bandit OMD within-task`
    $\mathbf{x}_{t+1} \leftarrow \frac{1}{t} \sum_{s=1}^t \hat{\mathbf{x}}_s$             `// FTL update of initialization`
    **for** $\theta \in \Theta_k$ **do**
        $\eta_{t+1}(\theta) \leftarrow \frac{\int_{\underline{\eta}(\theta)}^{\overline{\eta}(\theta)} v \exp\left(-\alpha(\theta) \sum_{s=1}^t U_s^{(\rho)}(\mathbf{x}_s, v, \theta)\right) dv}{\int_{\underline{\eta}(\theta)}^{\overline{\eta}(\theta)} \exp\left(-\alpha(\theta) \sum_{s=1}^t U_s^{(\rho)}(\mathbf{x}_s, v, \theta)\right) dv}$    `// EWOO step-size update`
        $\mathbf{w}_{t+1}(\theta) \leftarrow \mathbf{w}_t(\theta) - \lambda U_t(\mathbf{x}_t, \eta_t(\theta), \theta)$ `// MW update of tuning parameter`

---

OMD on **loss estimators** $\hat{\ell}_{t,i}$ constructed via partial feedback forms an important class of bandit methods [6, 2, 3]. Their regularizers $\psi$ are often non-Lipschitz, e.g. the negative entropy, or even unbounded, e.g. the log-barrier. Thus full-information results for tuning OMD, e.g. by Khodak et al. [31] and Denevi et al. [19], do not suffice. We do adapt the former's approach of online learning a sequence $U_t(\mathbf{x}, \eta, \theta)$ of affine functions of Bregman divergences from initializations $\mathbf{x}$ to known points in $\mathcal{K}$. We are interested in them because the regret of OMD w.r.t. a comparator $\mathbf{y}$ is bounded by $B(\mathbf{y}||\mathbf{x})/\eta + \mathcal{O}(\eta m)$ [46, 25]. In our case the comparator is based on the estimated optimum $\hat{\mathbf{x}}_t \in \arg\min_{\mathbf{x} \in \mathcal{K}} \langle \hat{\ell}_t, \mathbf{x} \rangle$, where $\hat{\ell}_t = \sum_{i=1}^m \hat{\ell}_{t,i}$, resulting from running OMD on task $t$ using initialization $\mathbf{x} \in \mathcal{K}$ and hyperparameters $\eta$ and $\theta$, which we denote $\text{OMD}_{\eta,\theta}(\mathbf{x})$. Unlike full-information meta-learning, we use a parameter $\varepsilon > 0$ to constrain this optimum to lie in a subset $\mathcal{K}_\varepsilon \subset \mathcal{K}^\circ$. Formally, we fix a point $\mathbf{x}_1 \in \mathcal{K}^\circ$ to be the "center"—e.g. $\mathbf{x}_1 = \mathbf{1}_d/d$ when $\mathcal{K}$ is the $d$-simplex $\triangle$—and define the projection $\mathbf{c}_\varepsilon(\mathbf{x}) = \mathbf{x}_1 + \frac{\mathbf{x}-\mathbf{x}_1}{1+\varepsilon}$ mapping from $\mathcal{K}$ to $\mathcal{K}_\varepsilon$. For example, $\mathbf{c}_{\frac{\varepsilon}{1-\varepsilon}}(\mathbf{x}) = (1-\varepsilon)\mathbf{x} + \varepsilon\mathbf{1}_d/d$ on the simplex. This projection allows us to handle regularizers $\psi$ that diverge near the boundary, but also introduces $\varepsilon$-dependent error terms. In the BLO case it also forces us to tune $\varepsilon$ itself, as initializing too close to the boundary leads to unbounded regret while initializing too far away does not take advantage of task-similarity. Thus the general upper bounds of interest are the following functions of the initialization $\mathbf{x}$, the step-size $\eta > 0$, and a third parameter $\theta$ that is either $\beta$ or $\varepsilon$, depending on the setting (MAB or BLO):

$$U_t(\mathbf{x}, \eta, \theta) = \frac{B_\theta(\mathbf{c}_\theta(\hat{\mathbf{x}}_t)||\mathbf{x})}{\eta} + \eta g(\theta)m + f(\theta)m \tag{3}$$

Here $B_\theta$ is the Bregman divergence of $\psi_\theta$ while $g(\theta) \geq 1$ and $f(\theta) \geq 0$ are tunable constants. We overload $\theta$ to be either $\beta$ or $\varepsilon$ for notational simplicity, as we will not tune them simultaneously; if $\theta = \beta$ (for MAB) then $c_\theta(\mathbf{x}) = \mathbf{x}_1 + \frac{\mathbf{x}-\mathbf{x}_1}{1+\varepsilon}$ for fixed $\varepsilon$, while if $\theta = \varepsilon$ (for BLO) then $B_\theta$ is the Bregman divergence of a fixed $\psi$. The reason to optimize this sequence of upper bounds $U_t$ is because they directly bound the task-averaged regret while being no worse than the worst-case single-task regret. Furthermore, an average over Bregman divergences is minimized at the average $\hat{\bar{\mathbf{x}}} = \frac{1}{T} \sum_{t=1}^T \hat{\mathbf{x}}_t$, where it attains the value $\hat{V}_\theta^2 = \frac{1}{T} \sum_{t=1}^T \psi_\theta(\mathbf{c}_\theta(\hat{\mathbf{x}}_t)) - \psi_\theta(\mathbf{c}_\theta(\hat{\bar{\mathbf{x}}}))$ (c.f. Claim A.1). We will show that this quantity leads to intuitive and interpretable notions of task-similarity in all the applications we study.

## 2.2 A meta-algorithm for tuning bandit algorithms

To learn these functions $U_t(\mathbf{x}, \eta, \theta)$—and thus to meta-learn $\text{OMD}_{\eta,\theta}(\mathbf{x})$—our meta-algorithm sets $\mathbf{x}$ to be the projection $\mathbf{c}_\theta$ of the mean of the estimated optima—i.e. follow-the-leader (FTL) over the Bregman divergences in (3)—while simultaneously setting $\eta$ via EWOO and $\theta$ via discrete multiplicative weights (MW). We choose FTL, EWOO, and MW because each is well-suited to the way $U_t$ depends on $\mathbf{x}$, $\eta$, and $\theta$, respectively. First, the only effect of $\mathbf{x}$ on $U_t$ is via the Bregman

divergence $B_\theta(\mathbf{c}_\theta(\hat{\mathbf{x}}_t)||\mathbf{x})$, over which FTL attains logarithmic regret [31]. For $\eta$, $U_t$ is exp-concave on $\eta > 0$ so long as the first term is nonzero, but it is also non-Lipschitz; the EWOO algorithm is one of the few methods with logarithmic regret on exp-concave losses without a dependence on the Lipschitz constant [26], and we ensure the first term is nonzero by *regularizing* the upper bounds as follows for some $\rho > 0$ and $D_\theta^2 = \max_{\mathbf{x},\mathbf{y} \in \mathcal{K}_\theta} B_\theta(\mathbf{x}||\mathbf{y})$:

$$U_t^{(\rho)}(\mathbf{x}, \eta, \theta) = \frac{B_\theta(\mathbf{c}_\theta(\hat{\mathbf{x}}_t)||\mathbf{x}) + \rho^2 D_\theta^2}{\eta} + \eta g(\theta)m + f(\theta)m \tag{4}$$

Note that this function is fully defined after obtaining $\hat{\mathbf{x}}_t$ by running OMD on task $t$, which allows us to use full-information MW to tune $\theta$ across the grid $\Theta_k$. Showing low regret w.r.t. any $\theta \in \Theta \supset \Theta_k$ then just requires sufficiently large $k$ and Lipschitzness of $U_t$ w.r.t. $\theta$. Combining all three algorithms together thus yields the guarantee in Theorem 2.1, which is our main structural result. It implies a generic approach for obtaining meta-learning algorithms by (1) bounding the task-averaged regret by an average of functions of the form $U_t$, (2) applying the theorem to obtain a new bound $o_T(1) + \min_{\theta,\eta} \frac{\hat{V}_\theta^2}{\eta} + \eta g(\theta)m + f(\theta)m$, and (3) bounding the estimated task-similarity $\hat{V}_\theta^2$ by an interpretable quantity. Crucially, since we can choose any $\eta > 0$, the asymptotic regret is always as good as the worst-case guarantee for running the base-learner separately on each task.

**Theorem 2.1** (c.f. Thm. A.1). *Suppose* $\mathbf{x}_1 = \arg\min_{\mathbf{x} \in \mathcal{K}} \psi_\theta(\mathbf{x}) \; \forall \; \theta$ *and let* $D$, $M$, $F$, *and* $S$ *be maxima over* $\theta$ *of* $D_\theta$, $D_\theta \sqrt{g(\theta)m}$, $f(\theta)$, *and* $\|\nabla^2 \psi_\theta\|_2$, *respectively. For each* $\rho \in (0,1)$ *we can set* $\underline{\eta}$, $\bar{\eta}$, $\alpha$, *and* $\lambda$ *s.t. the expected average of the losses* $U_t(\mathbf{c}_{\theta_t}(\mathbf{x}_t), \eta_t(\theta_t), \theta_t)$ *of Algorithm 1 is at most*

$$\min_{\theta \in \Theta, \eta > 0} \frac{\mathbb{E}\hat{V}_\theta^2}{\eta} + \eta g(\theta)m + f(\theta)m + \tilde{\mathcal{O}}\left( \frac{\frac{M}{\rho} + Fm}{\sqrt{T}} + \frac{L_\eta}{k} + \frac{M}{\rho^2 T} + \min\left\{\frac{\rho^2 D^2}{\eta}, \rho M\right\} + \frac{S}{\eta T} \right) \tag{5}$$

*Here* $\hat{V}_\theta^2 = \frac{1}{T}\sum_{t=1}^T \psi_\theta(\mathbf{c}_\theta(\hat{\mathbf{x}}_t)) - \psi_\theta(\mathbf{c}_\theta(\hat{\bar{\mathbf{x}}}))$ *and* $L_\eta$ *bounds the Lipschitz constant w.r.t.* $\theta$ *at* $\hat{V}_\theta^2/\eta + \eta g(\theta)m + f(\theta)m$. *The same bound plus* $(M/\rho + Fm)\sqrt{\frac{1}{T}\log\frac{1}{\delta}}$ *holds w.p.* $\geq 1 - \delta$.

We keep details of the dependence on $S$ and other constants as they are important in applying this result, but in most cases setting $\rho = \frac{1}{\sqrt[4]{T}}$ yields $\tilde{\mathcal{O}}(T^{\frac{3}{4}})$ regret. While a slow rate, the losses $U_t$ are non-Lipschitz and non-convex in-general, and learning them allows us to tune $\theta$ over user-specified intervals and $\eta$ over all positive numbers, which will be crucial later. At the same time, this tuning is what leads to the slow rate, as without tuning ($k = 1$, $L_\eta = 0$) the same $\rho$ yields $\tilde{\mathcal{O}}(\sqrt{T})$ regret. Lastly, while we focus on learning guarantees, we note that Algorithm 1 is reasonably efficient, requiring a $2k$ single-dimensional integrals per task; this is discussed in more detail in Section A.3.

## 3 Multi-armed bandits

We now turn to our first application: the multi-armed bandit problem, where at each round $i$ of task $t$ we take action $a_{t,i} \in [d]$ and observe loss $\ell_{t,i}(a_{t,i}) \in [0,1]$. As we are sampling actions from distributions $\mathbf{x} \in \mathcal{K} = \triangle$ on the $k$-simplex, the inner product $\langle \ell_{t,i}, \mathbf{x}_{t,i} \rangle$ is the expected loss and the optimal arm $\mathring{a}_t$ on task $t$ can be encoded as a vector $\mathring{\mathbf{x}}_t$ s.t. $\mathring{\mathbf{x}}_t(a) = 1_{a=\mathring{a}_t}$.

We use as a base-learner a generalization of Exp3 that uses the negative Tsallis entropy $\psi_\beta(\mathbf{p}) = \frac{1 - \sum_{a=1}^d \mathbf{p}^\beta(a)}{1 - \beta}$ for some $\beta \in (0,1]$ as the regularizer; this improves regret from Exp3's $\mathcal{O}(\sqrt{dm \log d})$ to the optimal $\mathcal{O}(\sqrt{dm})$ [3]. Note that $-\psi_\beta$ is the Shannon entropy in the limit $\beta \to 1$ and its Bregman divergence $B_\beta(\mathbf{x}||\cdot)$ is non-convex in the second argument. As the Tsallis entropy is non-Lipschitz at the simplex boundary, which is where the estimated and true optima $\hat{\mathbf{x}}_t$ and $\mathring{\mathbf{x}}_t$ lie, we will project them using $\mathbf{c}_{\frac{\varepsilon}{1-\varepsilon}}(\mathbf{x}) = (1-\varepsilon)\mathbf{x} + \varepsilon \mathbf{1}_d/d$ to the set $\mathcal{K}_{\frac{\varepsilon}{1-\varepsilon}} = \{\mathbf{x} \in \triangle : \min_a \mathbf{x}(a) \geq \varepsilon/d\}$. We denote the resulting vectors using the superscript $(\varepsilon)$, e.g. $\hat{\mathbf{x}}_t^{(\varepsilon)} = \mathbf{c}_{\frac{\varepsilon}{1-\varepsilon}}(\hat{\mathbf{x}}_t)$, and also use $\triangle^{(\varepsilon)} = \mathcal{K}_{\frac{\varepsilon}{1-\varepsilon}}$ to denote the constrained simplex. For MAB we also study two base-learners: (1) **implicit exploration** and (2) **guaranteed exploration**. The former uses low-variance loss *under*-estimators $\hat{\ell}_{t,i}(a) = \frac{\ell_{t,i}(a) 1_{a_{t,i}=a}}{\mathbf{x}_{t,i}(a) + \gamma}$ for $\gamma > 0$, where $\mathbf{x}_{t,i}(a)$ is the probability of sampling $a$ on task $t$ round $i$, to enable high probability bounds [43]. On the other hand, **guaranteed exploration** uses unbiased loss estimators (i.e. $\gamma = 0$) but constrains the action space to $\triangle^{(\varepsilon)}$, which we will use to adapt to a task-similarity determined by the *true* optima-in-hindsight.

## 3.1 Adapting to low estimated entropy with high probability using implicit exploration

In our first setting, the base-learner runs $\mathrm{OMD}_{\eta_t,\beta_t}(\mathbf{x}_{t,1})$ on $\gamma$-regularized estimators with Tsallis regularizer $\psi_{\beta_t}$, step-size $\eta_t$, and initialization $\mathbf{x}_{t,1} \in \triangle^{(\varepsilon)}$. Standard OMD analysis combined with implicit exploration analysis [43] shows (44) that the task-averaged regret is bounded w.h.p. by

$$(\varepsilon + \gamma d)m + \tilde{\mathcal{O}}\left(\frac{\sqrt{d}}{\gamma T}\right) + \frac{1}{T}\sum_{t=1}^{T} \frac{B_{\beta_t}(\hat{\mathbf{x}}_t^{(\varepsilon)}\|\mathbf{x}_{t,1})}{\eta_t} + \frac{\eta_t d^{\beta_t} m}{\beta_t} \tag{6}$$

The summands have the desired form of $U_t(\mathbf{x}_{t,1}, \eta_t, \beta_t)$, so by Theorem 2.1 we can bound their average by

$$\min_{\beta \in [\underline{\beta}, \overline{\beta}], \eta > 0} \frac{\hat{V}_\beta^2}{\eta} + \frac{\eta d^\beta m}{\beta} + \tilde{\mathcal{O}}\left(\frac{L_\eta}{k} + \frac{\left(\frac{d}{\varepsilon}\right)^{2-\beta}}{\eta T} + \left(\rho + \frac{1}{\rho\sqrt{T}} + \frac{1}{\rho^2 T}\right)d\sqrt{m}\right) \tag{7}$$

where $\hat{V}_\beta^2 = \frac{1}{T}\sum_{t=1}^{T}\psi_\beta(\hat{\mathbf{x}}_t^{(\varepsilon)}) - \psi_\beta(\hat{\bar{\mathbf{x}}}^{(\varepsilon)})$ is the average difference in Tsallis entropies between the ($\varepsilon$-constrained) estimated optima $\hat{\mathbf{x}}_t$ and their empirical distribution $\hat{\bar{\mathbf{x}}} = \frac{1}{T}\sum_{t=1}^{T}\hat{\mathbf{x}}_t$, while $L_\eta$ is the Lipschitz constant of $\frac{\hat{V}_\beta^2}{\eta} + \frac{\eta d^\beta m}{\beta}$ w.r.t. $\beta \in [\underline{\beta}, \overline{\beta}]$. The specific instantiation of Algorithm 1 that (7) holds for is to do the following at each time $t$:

1. sample $\beta_t$ via the MW distribution $\propto \exp(\mathbf{w}_t)$ over the discretization $\Theta_k$ of $[\underline{\beta}, \overline{\beta}] \subset [0,1]$

2. run $\mathrm{OMD}_{\eta_t,\beta_t}$ using the initialization $\mathbf{x}_{t,1} = \frac{1}{t-1}\sum_{s<t}\hat{\mathbf{x}}_t^{(\varepsilon)} = \frac{\varepsilon}{d}\mathbf{1}_d + \frac{1-\varepsilon}{t-1}\sum_{s<t}\hat{\mathbf{x}}_t$ (FTL)

3. update EWOO at each $\beta \in \Theta_k$ with loss $\frac{B_\beta(\hat{\mathbf{x}}_t^{(\varepsilon)}\|\mathbf{x}_{t,1})+\rho^2 D_\beta^2}{\eta} + \frac{\eta d^\beta m}{\beta}$, where $D_\beta^2 = \frac{d^{1-\beta}-1}{1-\beta}$

4. update $\mathbf{p}_{t+1}$ using multiplicative weights with expert losses $\frac{B_\beta(\hat{\mathbf{x}}_t^{(\varepsilon)}\|\mathbf{x}_{t,1})}{\eta} + \frac{\eta d^\beta m}{\beta}$

$$\tag{8}$$

The final guarantee for this procedure, given in full in Theorem B.1, follows by two properties of the Tsallis entropy $-\psi_\beta$: (1) its Lipschitzness w.r.t. $\beta \in [0,1]$ (c.f. Lem B.1) and (2) the fact that $\hat{V}_\beta^2$ is bounded by the entropy $\hat{H}_\beta = -\psi_\beta(\hat{\bar{\mathbf{x}}})$ of the empirical distribution of estimated optima (c.f. Lem B.2), which yields our first notion of task-similarity: *multi-armed bandit tasks are similar if the empirical distribution of their (estimated) optimal arms has low entropy.*

We exemplify the implications of Theorem B.1 in Corollary 3.1, where we consider three regimes of the lower bound $\underline{\beta}$ on the entropy parameter: $\underline{\beta} = 1$, i.e. always using Exp3; $\underline{\beta} = 1/2$, which corresponds to the optimal worst-case setting [3]; and $\underline{\beta} = 1/\log d$, below which the OMD regret-upper-bound always worsens (and so it does not make sense to try $\beta < 1/\log d$).

**Corollary 3.1** (c.f. Cors. B.1, B.2, and B.3). *Suppose $\overline{\beta} = 1$ and we set the initialization, step-size, and entropy parameter of Tsallis OMD with implicit exploration via Algorithm 1 as in Theorem B.1.*

*1. If $\underline{\beta} = 1$ and $T \geq \frac{d^2}{m}$ we can ensure $\frac{1}{T}\sum_{t=1}^{T}\sum_{i=1}^{m}\ell_{t,i}(\mathbf{x}_{t,i}) - \ell_{t,i}(\mathring{\mathbf{x}}_t) \leq 2\sqrt{\hat{H}_1 dm} + \tilde{\mathcal{O}}\left(\frac{d^{\frac{2}{3}}m^{\frac{2}{3}}}{\sqrt[3]{T}}\right)$ w.h.p.*

*2. If $\underline{\beta} = \frac{1}{2}$ and $T \geq \frac{d^{5/2}}{m}$ we can set $k = \lceil\sqrt[4]{d}\sqrt{T}\rceil$ and ensure w.h.p. that task-averaged regret is*

$$\min_{\beta \in [\frac{1}{2},1]} 2\sqrt{\hat{H}_\beta d^\beta m/\beta} + \tilde{\mathcal{O}}\left(\frac{d^{5/7}m^{5/7}}{T^{2/7}} + \frac{d\sqrt{m}}{\sqrt[4]{T}}\right) \tag{9}$$

*3. If $\underline{\beta} = \frac{1}{\log d}$ and $T \geq \frac{d^3}{m}$ we can set $k = \lceil\sqrt[4]{d}\sqrt{T}\rceil$ and ensure w.h.p. that task-averaged regret is*

$$\min_{\beta \in (0,1]} 2\sqrt{\hat{H}_\beta d^\beta m/\beta} + \tilde{\mathcal{O}}\left(\frac{d^{3/4}m^{3/4} + d\sqrt{m}}{\sqrt[4]{T}}\right) \tag{10}$$

In all three settings, as $T \to \infty$ the regret scales directly with the entropy of the estimated optima-in-hindsight, which is small if most tasks are estimated to be solved by one of a few arms and large if all arms are used roughly equally. Corollary 3.1 demonstrates the importance of tuning $\beta$: even if tasks are dissimilar, we asymptotically recover the worst-case optimal guarantee $\mathcal{O}(\sqrt{dm})$ in cases two and three because the entropy is at most $\frac{d^{1-\beta}}{1-\beta}$. On the other hand, if a constant $s \ll d$ actions are always minimizers, i.e. the empirical distribution $\hat{\bar{\mathbf{x}}}$ is $s$-sparse, then the last bound (10)

implies that Algorithm 1 can achieve task-averaged regret $o_T(md) + \mathcal{O}(\sqrt{sm \log d})$. At the same time, this tuning is costly, with the last two results having an extra $\tilde{\mathcal{O}}\left(\frac{d\sqrt{m}}{\sqrt[4]{T}}\right)$ term because of it. Furthermore, the bound of $\beta = \frac{1}{2}$ has a slightly better dependence on $d$, $m$, and $T$ compared to that of $\underline{\beta} = \frac{1}{\log d}$ due to the $\left(\frac{d}{\varepsilon}\right)^{2-\beta}$ term in the bound (7) returned for MAB by our structural result.

We can compare the $s$-sparse result to Azizi et al. [10], who achieve task-averaged regret $\tilde{\mathcal{O}}(m/\sqrt[3]{T} + \sqrt{sm \log T})$ for *stochastic* MAB. Despite our adversarial setting and no stipulations on how tasks are related, our bounds are asymptotically comparable if the estimated and true optima are roughly equivalent (ignoring their $\mathcal{O}(\sqrt{\log T})$-factor), as we also have $\tilde{\mathcal{O}}(\sqrt{sm})$ average regret as $T \to \infty$. Their rate in the number of tasks is better, but at a cost of runtime exponential in $s$. Apart from generality, we believe a great strength of our results is their adaptiveness; unlike Azizi et al. [10], we do not need to know how many optimal arms there are to adapt to there being few of them.

## 3.2 Adapting to the entropy of the true optima-in-hindsight using guaranteed exploration

While the entropy of estimated optima-in-hindsight may be useful in some cases where we wish to actually *compute* the task-similarity, it is otherwise generally more desirable to adapt to an intrinsic and algorithm-independent measure, e.g. the entropy of the *true* optima-in-hindsight. However, doing so is difficult without further assumptions, as the optima are both hard to identify and the measure itself may not be fully defined in case of ties. Thus in this section we focus on the setting where we have a nonzero performance gap $\Delta > 0$ between the best and second-best arms:

**Assumption 3.1.** *For some $\Delta > 0$ and all tasks $t \in [T]$, $\frac{1}{m} \sum_{i=1}^{m} \ell_{t,i}(a) - \ell_{t,i}(\mathring{a}_t) \geq \Delta \ \forall \ a \neq \mathring{a}_t$.*

This assumption is common in the best-arm identification literature [28, 1], which we adapt to show that the estimated optimal arms match the true optima, and thus so do their entropies. To do so, we switch to *unbiased* loss estimators, i.e. $\gamma = 0$, and control their variance by lower-bounding the probability of selecting an arm to be at least $\frac{\varepsilon}{d}$; this can alternatively be expressed as running OMD using the regularizer $\psi_\beta + I_{\triangle(\varepsilon)}$, where for any $\mathcal{C} \subset \mathbb{R}^d$ the function $I_\mathcal{C}(\mathbf{x}) = 0$ if $\mathbf{x} \in \mathcal{C}$ and $\infty$ otherwise. Guaranteed exploration allows us extend the analysis of Abbasi-Yadkori et al. [1] to show that the estimated arm is optimal w.h.p.:

**Lemma 3.1** (c.f. Lem C.1). *Suppose for $\varepsilon > 0$ and any $\beta \in (0, 1]$ we run OMD on task $t \in [T]$ with regularizer $\psi_\beta + I_{\triangle(\varepsilon)}$. If $m = \tilde{\Omega}(\frac{d}{\varepsilon\Delta^2})$ then $\hat{\mathbf{x}}_t = \mathring{\mathbf{x}}_t$ w.p. $\geq 1 - d \exp(-\Omega(\varepsilon\Delta^2 m/d))$.*

However, the constraint that the probabilities are at least $\frac{\varepsilon}{d}$ does lead to $\varepsilon m$ additional error on each task, with the upper bound on the task-averaged expected regret becoming

$$\mathbb{E} \frac{1}{T} \sum_{t=1}^{T} \sum_{i=1}^{m} \ell_{t,i}(a_{t,i}) - \ell_{t,i}(\mathring{a}_t) \leq \varepsilon m + \frac{1}{T} \sum_{t=1}^{T} \frac{\mathbb{E} B_{\beta_t}(\hat{\mathbf{x}}_t^{(\varepsilon)} \| \mathbf{x}_{t,1})}{\eta_t} + \frac{\eta_t d^{\beta_t} m}{\beta_t} \qquad (11)$$

Moreover, we will no longer set $\varepsilon = o_T(1)$, as this would require $m$ to be *increasing* in $T$ for the best-arm identification result of Lemma C.1 to hold. Thus, unlike in the previous section, our results will contain "fast" terms—terms in the task-averaged regret that are $o(\sqrt{m})$ but not decreasing in $T$ nor affected by the task-similarity. They will still improve upon the $\Omega(\sqrt{dm})$ MAB lower bound if tasks are similar, but the task-averaged regret will not converge to zero as $T \to \infty$ if the tasks are identical.

Nevertheless, the tuning-dependent component of the upper bounds in (11) has the appropriate form for our structural result—in fact we can use the same meta-algorithm (8) as for implicit exploration—and so we can again apply Theorem 2.1 to get a bound on the task-averaged regret in terms of the average difference $\hat{V}_\beta^2 = \frac{1}{T} \sum_{t=1}^{T} \psi_\beta(\hat{\mathbf{x}}_t^{(\varepsilon)}) - \psi_\beta(\bar{\hat{\mathbf{x}}}^{(\varepsilon)})$ of the entropies of the $\varepsilon$-constrained estimated task-optima $\hat{\mathbf{x}}_t^{(\varepsilon)}$ and their mean $\bar{\hat{\mathbf{x}}}^{(\varepsilon)}$. The easiest way to apply Lemma C.1 to bound $\hat{V}_\beta^2$ in terms of $H_\beta = \frac{1}{T} \sum_{t=1}^{T} \psi_\beta(\mathring{\mathbf{x}}_t) - \psi_\beta(\bar{\mathring{\mathbf{x}}})$ is via union bound on all $T$ tasks to show that $\hat{\mathbf{x}}_t = \mathring{\mathbf{x}}_t \ \forall \ t$ w.p. $\geq 1 - dT \exp(-\Omega(\varepsilon\Delta^2 m/d))$; however, setting a constant failure probability leads to $m$ growing, albeit only logarithmically, in $T$. Instead, by analyzing the worst-case best-arm identification probabilities, we show in Lemma C.2 that the expectation of $\hat{V}_\beta^2$ is bounded by $H_\beta + 3\beta \frac{(d/\varepsilon)^{1-\beta}-1}{1-\beta} \exp\left(-\frac{3\varepsilon\Delta^2 m}{28d}\right)$ without resorting to $m = \omega_T(1)$. Assuming $m \geq \frac{75d}{\varepsilon\Delta^2} \log \frac{d}{\varepsilon\Delta^2}$ is enough (69) to bound the second term by $\frac{56}{dm}$. Then the final result (c.f. Thm. C.1) bounds the expected task-averaged regret as follows (ignoring terms that become $o_T(1)$ after setting $\rho$ and $k$):

$$\varepsilon m + \min_{\beta \in [\underline{\beta}, \overline{\beta}], \eta > 0} \frac{h_\beta(\Delta)}{\eta} + \frac{\eta d^\beta m}{\beta} \quad \text{for} \quad h_\beta(\Delta) = \begin{cases} H_\beta + \frac{56}{md} & \text{if} \quad m \geq \frac{75d}{\varepsilon \Delta^2} \log \frac{d}{\varepsilon \Delta^2} \\ \frac{d^{1-\beta}-1}{1-\beta} & \text{otherwise} \end{cases} \quad (12)$$

If the gap $\Delta$ is known and sufficiently large, then we can set $\varepsilon = \Theta(\frac{d}{\Delta^2 m})$ to obtain an asymptotic task-averaged regret that scales only with the entropy $H_\beta$ and a fast term that is logarithmic in $m$:

**Corollary 3.2** (c.f. Cor. C.3). *Suppose we set the initialization, step-size, and entropy parameter of Tsallis OMD with guaranteed exploration via Algorithm 1 as specified in Theorem C.1. If $[\underline{\beta}, \overline{\beta}] = [\frac{1}{\log d}, 1]$ and $m \geq \frac{75d}{\Delta^2} \log \frac{d}{\Delta^2}$, then setting $\varepsilon = \tilde{\Theta}\left(\frac{d}{\Delta^2 m}\right)$, $\rho = \frac{1}{\sqrt[3]{d} \sqrt[6]{mT}}$, and $k = \lceil \sqrt[3]{d^2 mT} \rceil$ ensures that the expected task-averaged regret is at most*

$$\min_{\beta \in (0,1]} 2\sqrt{H_\beta d^\beta m/\beta} + \tilde{\mathcal{O}}\left(\frac{d}{\Delta^2} + \frac{d^{\frac{4}{3}} m^{\frac{2}{3}}}{\sqrt[3]{T}} + \frac{d^{\frac{5}{3}} m^{\frac{5}{6}}}{T^{\frac{2}{3}}} + \frac{d\Delta^4 m^3}{T}\right) \quad (13)$$

Knowing the gap $\Delta$ is a strong assumption, as ideally we could set $\varepsilon$ without it. Note that if $\varepsilon = \Omega(\frac{1}{m^p})$ for some $p \in (0, 1)$ then the condition $m \geq \frac{75d}{\varepsilon \Delta^2} \log \frac{d}{\varepsilon \Delta^2}$ only fails if $m \leq \text{poly}(\frac{1}{\Delta})$, i.e. for gap decreasing in $m$. We can use this together with the fact that minimizing over $\eta$ and $\beta$ in our bound allows us to replace them with any value, even a gap-dependent one, to derive a gap-*independent* setting of $\varepsilon$ that ensures a task-similarity-adaptive bound when $\Delta$ is not too small and falls back to the worst-case optimal guarantee otherwise. Specifically, for indicator $\iota_\Delta = 1_{m \geq \frac{75d}{\varepsilon \Delta^2} \log \frac{d}{\varepsilon \Delta^2}}$, setting $\eta = \Theta\left(\sqrt{\frac{h_\beta(\Delta)}{d^\beta m/\beta}}\right)$ in (12) and using $\beta = \frac{1}{2}$ if the condition $\iota_\Delta$ fails yields asymptotic regret at most

$$\varepsilon m + \min_{\beta \in (0,1]} \mathcal{O}\left(\iota_\Delta \sqrt{\frac{H_\beta d^\beta m}{\beta}} + (1 - \iota_\Delta)\sqrt{dm}\right) \leq \varepsilon m + \tilde{\mathcal{O}}\left(\min\left\{\min_{\beta \in (0,1]} \sqrt{\frac{H_\beta d^\beta m}{\beta}} + \frac{d}{\Delta\sqrt{\varepsilon}}, \sqrt{dm}\right\}\right) \quad (14)$$

Thus setting $\varepsilon = \Theta(\sqrt{d}/m^{\frac{2}{3}})$ yields the desired dependence on the entropy $H_\beta$ and a fast term in $m$:

**Corollary 3.3** (c.f. Cor. C.4). *In the setting of Corollary 3.2 but with $m = \Omega(d^{\frac{3}{4}})$ and unknown $\Delta$, using $\varepsilon = \Theta(\sqrt{d}/m^{\frac{2}{3}})$ ensures expected task-averaged regret at most*

$$\min\left\{\min_{\beta \in (0,1]} 2\sqrt{H_\beta d^\beta m/\beta} + \tilde{\mathcal{O}}\left(\frac{d^{\frac{3}{4}} \sqrt[3]{m}}{\Delta}\right), 8\sqrt{dm}\right\} + \tilde{\mathcal{O}}\left(\frac{d^{\frac{4}{3}} m^{\frac{2}{3}}}{\sqrt[3]{T}} + \frac{d^{\frac{5}{3}} m^{\frac{5}{6}}}{T^{\frac{2}{3}}} + \frac{d^2 m^{\frac{7}{3}}}{T}\right) \quad (15)$$

While not logarithmic, the gap-dependent term is still $o(\sqrt{m})$, and moreover the asymptotic regret is no worse than the worst-case optimal $\mathcal{O}(\sqrt{dm})$. Note that the latter is only needed if $\Delta = o(1/\sqrt[6]{m})$.

The main improvement in this section is in using the entropy of the true optima, which can be much smaller than that of the estimated optima if there are a few good arms but large noise. Our use of the gap assumption for this seems difficult to avoid for this notion of task-similarity. We can also compare to Corollary 3.1 (10), which did not require $\Delta > 0$ and had no fast terms but had a worse rate in $T$; in contrast, the $\mathcal{O}(\frac{1}{\sqrt[3]{T}})$ rates above match that of the closest stochastic bandit result [10]. As before, for $s \ll d$ "good" arms we obtain $\mathcal{O}(\sqrt{sm \log d})$ asymptotic regret, assuming the gap is not too small. Finally, we can also compare to the classic shifting regret bound for Exp3.S [6], which translated to task-averaged regret is $\mathcal{O}(\sqrt{dm \log(dmT)})$. This is worse than even running OMD separately on each task, albeit under weaker assumptions (not knowing task boundaries). It also cannot take advantage of repeated optimal arms, e.g. the case of $s \ll d$ good arms.

### 3.3 Adapting to entropic task similarity implies robustness to outliers

While we considered mainly the $s$-sparse setting as a way of exemplifying our results and comparing to other work such as Azizi et al. [10], the fact that our approach can adapt to the Tsallis entropy $\min_\beta H_\beta$ of the optimal arms implies meaningful guarantees for any low-entropy distribution over the optimal arms, not just sparsely-supported ones. One way to illustrate the importance of this is through an analysis of robustness to outlier tasks. Specifically, suppose that the $s$-sparsity assumption—that optima $\mathring{a}_t$ lie in a subset of $[T]$ of size $s \ll d$—only holds for all but $\mathcal{O}(T^p)$ of the tasks $t \in [T]$, where $p \in [0, 1)$. Then the best we can do using an asymptotic bound of $\tilde{\mathcal{O}}(\sqrt{sm})$—e.g. that of Azizi et al.

[10] in the stochastic case or from naively applying $\min_{\beta \in (0,1]} H_\beta d^\beta m / \beta \leq esm \log d$ to any of our previous results—is to substitute $s + T^p$ instead of $s$, which will only improve over the single-task bound if $d = \omega(T^p)$, i.e. in the regime where the number of arms increases with the number of tasks.

However, our notion of task-similarity allows us to do much better, as we can show (c.f. Prop. D.1) that in the same setting $H_\beta = \mathcal{O}(s + \frac{d^{1-\beta}}{T^{\beta(1-p)}})$ for any $\beta \in [\frac{1}{\log d}, \frac{1}{2}]$. Substituting this result into e.g. Corollary 3.3 yields the same asymptotic result of $\mathcal{O}(\sqrt{sm \log d})$, although the rate in $T$ is a very slow $\mathcal{O}(\sqrt{dm}/T^{\frac{1-p}{2\log d}})$. This demonstrates how our entropic notion of task-similarity simultaneously yields strong results in the $s$-sparse setting and is meaningful in more general settings.

## 4 Bandit linear optimization

Our last application is bandit linear optimization, in which at task $t$ round $i$ we play $\mathbf{x}_{t,i} \in \mathcal{K}$ in some convex $\mathcal{K} \subset \mathbb{R}^d$ and observe loss $\langle \ell_{t,i}, \mathbf{x}_{t,i} \rangle \in [-1, 1]$. We will again use a variant of mirror descent, using a **self-concordant barrier** for $\psi$ and the specialized loss estimators of Abernethy et al. [2, Alg. 1]. More information on such regularizers can be found in the literature on interior point methods [42]. We pick this class of algorithms because of their optimal dependence on the number of rounds and their applicability to any convex domain $\mathcal{K}$ via specific barriers $\psi$, which will yield interesting notions of task-similarity. Our ability to handle non-smooth regularizers via the structural result (Thm. 2.1) is even more important here, as barriers are infinite at the boundaries. Indeed, we will *not* learn a $\beta$ parameterizing the regularizer and instead focus on tuning a boundary offset $\varepsilon > 0$. Here we make use of notation from Section 2, where $\mathbf{c}_\varepsilon$ maps points in $\mathcal{K}$ to a subset $\mathcal{K}_\varepsilon$ defined by the Minkowski function (c.f. Def. E.1) centered at $\mathbf{x}_1 = \arg\min_{\mathbf{x} \in \mathcal{K}} \psi(\mathbf{x})$.

From Abernethy et al. [2] we have an upper bound on the expected task-averaged regret of their algorithm run from initializations $\mathbf{x}_{t,1} \in \mathcal{K}^\circ$ with step-sizes $\eta_t > 0$ and offsets $\varepsilon_t > 0$:

$$\mathbb{E}\frac{1}{T}\sum_{t=1}^{T}\sum_{i=1}^{m}\langle \ell_{t,i}, \mathbf{x}_{t,i} - \mathring{\mathbf{x}}_t \rangle \leq \frac{1}{T}\sum_{t=1}^{T}\frac{\mathbb{E}B(\mathbf{c}_{\varepsilon_t}(\hat{\mathbf{x}}_t)||\mathbf{x}_{t,1})}{\eta_t} + (32\eta_t d^2 + \varepsilon_t)m \qquad (16)$$

We can show (88) that $D_\varepsilon^2 = \max_{\mathbf{x}, \mathbf{y} \in \mathcal{K}_\varepsilon} B(\mathbf{x}||\mathbf{y}) \leq \frac{9\nu^{\frac{3}{2}} K \sqrt{S_1}}{\varepsilon}$, where $\nu$ is the self-concordance constant of $\psi$ and $S_1 = \|\nabla^2 \psi(\mathbf{x}_1)\|_2$ is the spectral norm of its Hessian at the center $\mathbf{x}_1$ of $\mathcal{K}$. Restricting to tuning $\varepsilon \in [\frac{1}{m}, 1]$—which is enough to obtain constant task-averaged regret above if the estimated optima $\hat{\mathbf{x}}_t$ are identical—we can now apply Algorithm 1 via the following instantiation:

1. sample $\varepsilon_t$ via the MW distribution $\propto \exp(\mathbf{w}_t)$ over the discretization $\Theta_k$ of $[\frac{1}{m}, 1]$

2. run $\mathtt{OMD}_{\eta_t, \varepsilon_t}$ using the initialization $\mathbf{x}_{t,1} = \frac{1}{t-1}\sum_{s<t}\mathbf{c}_{\varepsilon_t}(\hat{\mathbf{x}}_t) = \mathbf{x}_1 + \frac{\sum_{s<t}\hat{\mathbf{x}}_t - \mathbf{x}_1}{(1+\varepsilon_t)(t-1)}$ (FTL)

3. update EWOO at each $\varepsilon \in \Theta_k$ with loss $\frac{B(\mathbf{c}_\varepsilon(\hat{\mathbf{x}}_t)||\mathbf{x}_{t,1}) + \rho^2 D_\varepsilon^2}{\eta} + 32\eta d^2$ for $D_\varepsilon^2 = \frac{9\nu^{\frac{3}{2}} K \sqrt{S_1}}{\varepsilon}$

4. update $\mathbf{p}_{t+1}$ using multiplicative weights with expert losses $\frac{B(\mathbf{c}_\varepsilon(\hat{\mathbf{x}}_t)||\mathbf{x}_{t,1})}{\eta} + \varepsilon m$

$$\qquad (17)$$

Note the similarity to the MAB case (8), with the difference being the upper bound passed to EWOO and MW. Our structural result bounds the expected task-averaged regret as follows (c.f. Thm. E.1):

$$\mathbb{E}\min_{\varepsilon \in [\frac{1}{m}, 1], \eta > 0}\frac{\hat{V}_\varepsilon^2}{\eta} + (32\eta d^2 + \varepsilon)m + \tilde{\mathcal{O}}\left(\frac{\frac{m^2}{T} + \frac{1}{k}}{\eta} + \frac{m}{k} + m\min\left\{\frac{\rho^2}{\eta}, d\rho\right\} + \frac{dm}{\rho}\sqrt{\frac{\log k}{T}} + \frac{dm}{\rho^2 T}\right) \qquad (18)$$

For $\rho = o_T(1)$ and $k = \omega_T(1)$ this becomes $o_T(\text{poly}(m)) + \mathbb{E}\min_{\varepsilon \in [\frac{1}{m}, 1], \eta > 0}\frac{\hat{V}_\varepsilon^2}{\eta} + 32\eta d^2 m + \varepsilon m$, where $\hat{V}_\varepsilon^2 = \frac{1}{T}\sum_{t=1}^{T}\psi(\mathbf{c}_\varepsilon(\hat{\mathbf{x}}_t) - \psi(\mathbf{c}_\varepsilon(\hat{\mathbf{x}}_t))$. Then by tuning $\eta$ we get an asymptotic ($T \to \infty$) regret of $4d\hat{V}_\varepsilon\sqrt{2m} + \varepsilon m$ for any $\varepsilon \in [\frac{1}{m}, 1]$. Our analysis removes the explicit dependence on $\sqrt{\nu}$ that appears in the single-task regret [2]; as an example, $\nu$ equals the number of inequalities defining a polytope $\mathcal{K}$, as in the bandit shortest-path application below.

The remaining challenge is to interpret $\hat{V}_\varepsilon^2$, which as we did for MAB we do via specific examples, in this case concrete action domains $\mathcal{K}$. Our first example is for BLO over the unit sphere $\mathcal{K} = \{\mathbf{x} \in \mathbb{R}^d : \|\mathbf{x}\|_2 \leq 1\}$ using the appropriate log-barrier regularizer $\psi(\mathbf{x}) = -\log(1 - \|\mathbf{x}\|_2^2)$:

**Corollary 4.1** (c.f. Cor. E.1). *For BLO on the sphere, Algorithm 1 has expected task-averaged regret*

$$\tilde{\mathcal{O}}\left(\frac{dm^{\frac{3}{2}}}{T^{\frac{3}{4}}} + \frac{dm}{\sqrt[4]{T}}\right) + \min_{\varepsilon \in [\frac{1}{m}, 1]} 4d\sqrt{2m \log\left(1 + \frac{1 - \mathbb{E}\|\hat{\bar{\mathbf{x}}}\|_2^2}{2\varepsilon + \varepsilon^2}\right)} + \varepsilon m \qquad (19)$$

The bound above is decreasing in $\mathbb{E}\|\hat{\bar{\mathbf{x}}}\|_2^2$, the expected squared norm of the average of the estimated optima $\hat{\mathbf{x}}_t$. We thus say that *bandit linear optimization tasks over the sphere are similar if the norm of the empirical mean of their (estimated) optima is large*. This makes intuitive sense: if the tasks' optima are uniformly distributed, we should expect $\mathbb{E}\|\hat{\bar{\mathbf{x}}}\|_2^2$ to be small, even decreasing in $d$. On the other hand, in the degenerate case where the estimated optima $\hat{\mathbf{x}}_t$ are the same across all tasks $t \in [T]$, we have $\mathbb{E}\|\hat{\bar{\mathbf{x}}}\|_2^2 = 1$, so the asymptotic task-averaged regret is 1 because we can use $\varepsilon = \frac{1}{m}$. Perhaps slightly more realistically, if it is $\frac{1}{m^p}$-away from 1 for some power $p \geq \frac{1}{2}$ then setting $\varepsilon = \frac{1}{\sqrt{m}}$ can remove the logarithmic dependence on $m$. These two regimes illustrate the importance of tuning $\varepsilon$.

As a last application, we apply our meta-BLO result to the shortest-path problem in online optimization [50, 30]. In its bandit variant [8, 17], at each step $i = 1, \ldots, m$ the player must choose a path $p_i$ from a fixed source $u \in V$ to a fixed sink $v \in V$ in a directed graph $G(V, E)$. At the same time the adversary chooses edge-weights $\ell_i \in \mathbb{R}^{|E|}$ and the player suffers the sum $\sum_{e \in p_t} \ell_i(e)$ of the weights in their chosen path $p_t$. This can be relaxed as BLO over vectors $\mathbf{x}$ in a set $\mathcal{K} \subset [0, 1]^{|E|}$ defined by a set $\mathcal{C}$ of $\mathcal{O}(|E|)$ linear constraints $(\mathbf{a}, b) \langle \mathbf{a}, \mathbf{x} \rangle \leq b$ enforcing flows from $u$ to $v$; $u$ to $v$ paths can be sampled from any $\mathbf{x} \in \mathcal{K}$ in an unbiased manner [2, Proposition 1]. On a single-instance, applying the BLO method of Abernethy et al. [2] ensures $\mathcal{O}(|E|^{\frac{3}{2}}\sqrt{m})$ regret on this problem.

In the multi-instance setting, comprising a sequence $t = 1, \ldots, T$ of shortest path instances with $m$ adversarial edge-weight vectors $\ell_{t,i}$ each, we can attempt to achieve better performance by tuning the same method across instances. Notably, we can view this as the problem of learning predictions [33] in the algorithms with predictions paradigm from beyond-worst-case analysis [40], with the OMD initialization on each instance being effectively a prediction of its optimal path. Our meta-learner then has the following average performance across bandit shortest-path instances:

**Corollary 4.2** (c.f. Cor. E.2). *For multi-task bandit online shortest path, Algorithm 1 with regularizer* $\psi(\mathbf{x}) = -\sum_{\mathbf{a},b \in \mathcal{C}} \log(b - \langle \mathbf{a}, \mathbf{x} \rangle)$ *attains the following expected average regret across instances*

$$\tilde{\mathcal{O}}\left(\frac{|E|^4 m^{\frac{3}{2}}}{T^{\frac{3}{4}}} + \frac{|E|^{\frac{5}{2}} m^{\frac{5}{6}}}{\sqrt[4]{T}}\right) + \min_{\varepsilon \in [\frac{1}{m}, 1]} 4|E|\mathbb{E}\sqrt{2m \sum_{\mathbf{a},b \in \mathcal{C}} \log\left(\frac{\frac{1}{T}\sum_{t=1}^{T} b - \langle \mathbf{a}, \mathbf{c}_\varepsilon(\hat{\mathbf{x}}_t)\rangle}{\sqrt[T]{\prod_{t=1}^{T} b - \langle \mathbf{a}, \mathbf{c}_\varepsilon(\hat{\mathbf{x}}_t)\rangle}}\right)} + \varepsilon m \quad (20)$$

Here the asymptotic regret scales with the sum across all constraints $\mathbf{a}, b \in \mathcal{C}$ of the log of the ratio between the arithmetic and geometric means across tasks of the distances $b - \langle \mathbf{a}, \mathbf{c}_\varepsilon(\hat{\mathbf{x}}_t)\rangle$ from the estimated optimum flow $\mathbf{c}_\varepsilon(\hat{\mathbf{x}}_t)$ to the constraint boundary. As it is difficult to separate the effect of the offset $\varepsilon$, we do not state an explicit task-similarity measure like in our previous settings. Nevertheless, since the arithmetic and geometric means are equal exactly when all entries are equal—and otherwise the former is larger—the bound does show that regret is small when the estimated optimal flows $\hat{\mathbf{x}}_t$ for each task are at similar distances from the constraints, i.e. the boundaries of the polytope. Indeed, just as on the sphere, if the estimated optima are all the same then setting $\varepsilon = \frac{1}{m}$ again yields constant averaged regret.

# 5 Conclusion and limitations

We develop and apply a meta-algorithm for learning to initialize and tune bandit algorithms, obtaining task-averaged regret guarantees for both multi-armed and linear bandits that depend on natural, setting-specific notions of task similarity. For MAB, we meta-learn the initialization, step-size, and entropy parameter of Tsallis-entropic OMD and show good performance if the entropy of the optimal arms is small. For BLO, we use OMD with self-concordant regularizers and meta-learn the initialization, step-size, and boundary-offset, yielding interesting domain-specific task-similarity measures. Some natural directions for future work involve overcoming some limitations of our results: can we adapt to a notion of task-similarity that depends on the true optima without assuming a gap for MAB, or at all for BLO? Alternatively, can we design meta-learning algorithms that adapt to both stochastic and adversarial bandits, i.e. a "best-of-both-worlds" guarantee? Beyond this, one could explore other partial information settings, such as contextual bandits or bandit convex optimization.

## Acknowledgments

We thank our anonymous reviewers for helpful suggestions, especially concerning the analysis of robustness to outliers. This material is based on work supported in part by the National Science Foundation under grants CCF-1910321, FAI-1939606, IIS-1901403, SCC-1952085, and SES-1919453; the Defense Advanced Research Projects Agency under cooperative agreement HR00112020003; a Simons Investigator Award; an AWS Machine Learning Research Award; an Amazon Research Award; a Bloomberg Research Grant; a Microsoft Research Faculty Fellowship; a Google Faculty Research Award; a J.P. Morgan Faculty Award; a Facebook Research Award; a Mozilla Research Grant; a Facebook PhD Fellowship; and an NDSEG Fellowship. KYL is supported by the Israel Science Foundation (grant No. 447/20) and the Technion Center for Machine Learning and Intelligent Systems (MLIS). The work of RM was partially supported by the Israel Science Foundation grant number 1693/22.

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

# A  Structural results

## A.1  Properties of the Bregman divergence

**Lemma A.1.** *Let $\psi : \mathcal{C} \mapsto \mathbb{R}$ be a strictly convex function with $\max_{\mathbf{x} \in \mathcal{C}} \|\nabla^2 \psi(\mathbf{x})\|_2 \leq S$ over a convex set $\mathcal{C} \subset \mathbb{R}^d$ over size $\max_{\mathbf{x} \in \mathcal{C}} \|\mathbf{x}\|_2 \leq K$, and let $B(\cdot||\cdot)$ be the Bregman divergence generated by $\psi$. Then for any points $\mathbf{x}_1, \ldots, \mathbf{x}_T \in \mathcal{C}$ the actions $\mathbf{y}_1 = \arg\min_{\mathbf{x} \in \mathcal{C}} \psi(\mathbf{x})$ and $\mathbf{y}_t = \frac{1}{t-1} \sum_{s<t} \mathbf{x}_s$ have regret*

$$\sum_{t=1}^{T} B(\mathbf{x}_t || \mathbf{y}_t) - B(\mathbf{x}_t || \mathbf{y}_{T+1}) \leq \sum_{t=1}^{T} \frac{8SK^2}{2t-1} \leq 8SK^2 (1 + \log T) \tag{21}$$

*Proof.* Note that

$$\nabla_{\mathbf{y}} B(\mathbf{x}||\mathbf{y}) = -\nabla\psi(\mathbf{y}) - \nabla_{\mathbf{y}}\langle\nabla\psi(\mathbf{y}), \mathbf{x}\rangle + \nabla_{\mathbf{y}}\langle\nabla\psi(\mathbf{y}), \mathbf{y}\rangle = \text{diag}(\nabla^2\psi(\mathbf{y}))(\mathbf{y} - \mathbf{x}) \tag{22}$$

so $B(\mathbf{x}_t||\mathbf{y})$ is $2SK$-Lipschitz w.r.t. the Euclidean norm. Applying Khodak et al. [31, Prop. B.1] yields the result (note that its assumption of strong convexity of the regularizer can be replaced with strict convexity without changing the proof or result). □

**Claim A.1.** *Let $\psi : \mathcal{K} \mapsto \mathbb{R}$ be a strictly-convex function with Bregman divergence $B(\cdot||\cdot)$ over a convex set $\mathcal{K} \subset \mathbb{R}^d$ containing points $\mathbf{x}_1, \ldots, \mathbf{x}_T$. Then their mean $\bar{\mathbf{x}} = \frac{1}{T} \sum_{t=1}^{T} \mathbf{x}_t$ satisfies*

$$\sum_{t=1}^{T} B(\mathbf{x}_t || \bar{\mathbf{x}}) = \sum_{t=1}^{T} \psi(\mathbf{x}_t) - \psi(\bar{\mathbf{x}}) \tag{23}$$

*Proof.*

$$
\begin{aligned}
\sum_{t=1}^{T} B(\mathbf{x}_t || \bar{\mathbf{x}}) &= \sum_{t=1}^{T} \psi(\mathbf{x}_t) - \psi(\bar{\mathbf{x}}) - \langle\nabla\psi(\bar{\mathbf{x}}), \mathbf{x}_t - \bar{\mathbf{x}}\rangle \\
&= \sum_{t=1}^{T} \psi(\mathbf{x}_t) - \psi(\bar{\mathbf{x}}) - \langle\nabla\psi(\bar{\mathbf{x}}), \sum_{t=1}^{T}\mathbf{x}_t - \bar{\mathbf{x}}\rangle = \sum_{t=1}^{T} \psi(\mathbf{x}_t) - \psi(\bar{\mathbf{x}})
\end{aligned}
\tag{24}
$$

□

## A.2  Tuning the step-size

**Lemma A.2.** *Let $\ell_1, \ldots, \ell_T : \mathbb{R}_{>0} \mapsto \mathbb{R}_{>0}$ be a sequence of functions of form $\ell_t(x) = \frac{B_t^2}{x} + G^2 x$ for adversarially chosen $B_t \in [0, D]$ and some $G > 0$. Then for any $\rho \geq 0$, the actions of EWOO [26, Fig. 4] with parameter $\frac{2\rho^2}{DG}$ run on the modified losses $\frac{B_t^2 + \rho^2 D^2}{x} + G^2 x$ over the domain $\left[\frac{\rho D}{G}, \frac{D}{G}\sqrt{1 + \rho^2}\right]$ achieves regret w.r.t. any $x > 0$ of*

$$\sum_{t=1}^{T} \ell_t(x) - \ell_t(x) \leq \min\left\{\frac{\rho^2 D^2}{x}, \rho DG\right\} T + \frac{DG(1 + \log(T+1))}{2\rho^2} \tag{25}$$

*Proof.* By Khodak et al. [31, Prop. C.1] the modified functions are $\frac{2\rho^2}{DG}$-exp-concave. Then Khodak et al. [31, Cor. C.2] with $B_t$ set to $\frac{B_t}{G}$, $D$ to $\frac{D}{G}$, $\alpha_t = G^2$, and $\varepsilon = \frac{\rho D}{G}$ yields the result. □

**Lemma A.3.** *For* $\hat{\mathbf{x}}_1, \ldots, \hat{\mathbf{x}}_T \in \partial\mathcal{K}$ *consider a sequence of functions of form*

$$U_t(\mathbf{x}, \eta) = \frac{B(\mathbf{c}_\varepsilon(\hat{\mathbf{x}}_t)||\mathbf{x})}{\eta} + \eta G^2 m \tag{26}$$

*where* $B$ *is the Bregman divergence of a strictly convex d.g.f.* $\psi : \mathcal{K}^\circ \mapsto \mathbb{R}$ *and where* $\mathbf{x}_1 = \arg\min_{\mathbf{x} \in \mathcal{K}} \psi(\mathbf{x})$ *defines the projection* $\mathbf{c}_\varepsilon(\mathbf{x}) = \mathbf{x}_1 + \frac{\mathbf{x} - \mathbf{x}_1}{1+\varepsilon}$ *for some* $\varepsilon > 0$ . *Suppose we play* $\mathbf{x}_{t+1} \leftarrow \mathbf{c}_\varepsilon \left( \frac{1}{t} \sum_{s=1}^t \hat{\mathbf{x}}_s \right)$ *and set* $\eta_t$ *using the actions of EWOO [26, Fig. 4] with parameter* $\frac{2\rho^2}{DG}$ *for some* $\rho, D_\varepsilon > 0$ *s.t.* $B(\mathbf{c}_\varepsilon(\hat{\mathbf{x}}_t)||\mathbf{x}) \le D_\varepsilon^2 \, \forall \, \mathbf{x} \in \mathcal{K}_\varepsilon$ *on the functions* $\frac{B(\mathbf{c}_\varepsilon(\hat{\mathbf{x}}_t)||\mathbf{x}_t) + \rho^2 D_\varepsilon^2}{\eta} + \eta G^2 m$ *over the domain* $\left[ \frac{\rho D_\varepsilon}{G\sqrt{m}}, \frac{D_\varepsilon}{G}\sqrt{\frac{1+\rho^2}{m}} \right]$, *with* $\eta_1$ *being at the midpoint of the domain. Then* $U_t(\mathbf{x}_t, \eta_t) \le D_\varepsilon G \sqrt{m} \left( \frac{1}{\rho} + \sqrt{1+\rho^2} \right) \, \forall \, t \in [T]$ *and*

$$\sum_{t=1}^T U_t(\mathbf{x}_t, \eta_t) \le \min_{\eta > 0, \mathbf{x} \in \mathcal{K}} \sum_{t=1}^T \frac{B(\mathbf{c}_\varepsilon(\hat{\mathbf{x}}_t)||\mathbf{x})}{\eta} + \eta G^2 m$$
$$+ \min\left\{ \frac{\rho^2 D_\varepsilon^2}{\eta}, \rho D_\varepsilon G \right\} T + \frac{D_\varepsilon G(1 + \log(T+1))}{2\rho^2} + \frac{8 S_\varepsilon K^2 (1 + \log T)}{\eta} \tag{27}$$

*for* $K = \max_{\mathbf{x} \in \mathcal{K}} \|\mathbf{x}\|_2$ *and* $S_\varepsilon = \max_{\mathbf{x} \in \mathcal{K}_\varepsilon} \|\nabla^2 \psi(\mathbf{x})\|_2$.

*Proof.* The first claim follows by directly substituting the worst-case values of $\eta$ into $U_t(\mathbf{x}, \eta)$. For the second, apply Lemma A.2 followed by Lemma A.1:

$$\sum_{t=1}^T U_t(\mathbf{x}_t, \eta_t)$$
$$= \sum_{t=1}^T \frac{B(\mathbf{c}_\varepsilon(\hat{\mathbf{x}}_t)||\mathbf{x}_t)}{\eta_t} + \eta_t G^2 m$$
$$\le \min_{\eta > 0} \min\left\{ \frac{\rho^2 D_\varepsilon^2}{\eta}, \rho D_\varepsilon G \right\} T + \frac{D_\varepsilon G(1 + \log(T+1))}{2\rho^2} + \sum_{t=1}^T \frac{B(\mathbf{c}_\varepsilon(\hat{\mathbf{x}}_t)||\mathbf{x})}{\eta} + \eta G^2 m$$
$$\le \min_{\eta > 0} \min\left\{ \frac{\rho^2 D_\varepsilon^2}{\eta}, \rho D_\varepsilon G \right\} T + \frac{D_\varepsilon G(1 + \log(T+1))}{2\rho^2} + \frac{8 S_\varepsilon K^2 (1 + \log T)}{\eta}$$
$$+ \min_{\mathbf{x} \in \mathcal{K}_\varepsilon} \sum_{t=1}^T \frac{B(\mathbf{c}_\varepsilon(\hat{\mathbf{x}}_t)||\mathbf{x})}{\eta} + \eta G^2 m \tag{28}$$

Conclude by noting that the sum of Bregman divergence to $\mathbf{c}_\varepsilon(\hat{\mathbf{x}}_t)$ is minimized on their convex hull, a subset of $\mathcal{K}_\varepsilon$. $\qquad\square$

### A.3 Computational and space complexity

Algorithm 1 implicitly maintains a separate copy of FTL for each hyperparameter in the continuous space of EWOO and the grid $\Theta_k$ over the domain of $\theta$, but explicitly just needs to average the estimated task-optima $\hat{\mathbf{x}}_t$; this is due to the mean-as-minimizer property of Bregman divergences and the linearity of $\mathbf{c}_\varepsilon$. Thus the memory it uses is $\mathcal{O}(d+k)$, where $k$ is size of the discretization of $\Theta$ and should be viewed as sublinear in $T$, e.g. for MAB with implicit exploration and BLO $k = \mathcal{O}(\sqrt[4]{d}\sqrt{T})$. Computationally, at each timestep $t$ and for each grid point we must compute two single-dimensional integrals; the integrands are sums of upper bounds that just need to be incremented once per round, leading to a total per-iteration complexity of $\mathcal{O}(k)$ (ignoring the running of OMD). Although outside the scope of this work, it may be possible to avoid integration by tuning $\eta$ with MW as well, rather than EWOO, but likely at the cost of worse regret because it would not take advantage of the exp-concavity of $U_t^{(\rho)}$.

### A.4  Main structural result

**Theorem A.1.** *Consider a family of strictly convex functions $\psi_\theta : \mathcal{K}^\circ \mapsto \mathbb{R}$ parameterized by $\theta$ lying in an interval $\Theta \subset \mathbb{R}$ of radius $R_\Theta$ that are all minimized at the same $\mathbf{x}_1 \in \mathcal{K}^\circ$, and for $\hat{\mathbf{x}}_1, \ldots, \hat{\mathbf{x}}_T \in \partial\mathcal{K}$ consider a sequence of functions of form $U_t(\mathbf{x}, \eta, \theta)$ (3), as well as the associated regularized upper bounds $U_t^{(\rho)}$ (4). Define the maximum divergence $D = \max_{\theta \in \Theta} D_\theta$, radius $K = \max_{\mathbf{x} \in \mathcal{K}} \|\mathbf{x}\|_2$, and $L_\eta$ the Lipschitz constant w.r.t. $\theta \in \Theta$ of $\frac{\hat{V}_\theta^2}{\eta} + \eta g(\theta)m + f(\theta)m$. Then Algorithm 1 with $\Theta_k \subset \Theta$ the uniform discretization of $\Theta$ s.t. $\max_{\theta \in \Theta} \min_{\theta' \in \Theta_k} |\theta - \theta'| \le \frac{R_\Theta}{k}$, $\rho \in (0,1)$, $\underline{\eta}(\theta) = \frac{\rho D_\theta}{\sqrt{g(\theta)m}}$, $\overline{\eta}(\theta) = D_\theta \sqrt{\frac{1+\rho^2}{g(\theta)m}}$, $\alpha(\theta) = \frac{2\rho^2}{D_\theta\sqrt{g(\theta)m}}$, and $\lambda = \left( M\left(\frac{1}{\rho} + \sqrt{1+\rho^2}\right) + Fm \right)^{-1} \sqrt{\frac{\log k}{2T}}$ leads to a sequence $(\mathbf{x}_t, \eta_t(\theta_t), \theta_t)$ s.t. $\mathbb{E}\sum_{t=1}^T U_t(\mathbf{x}_t, \eta_t(\theta_t), \theta_t)$ is bounded by*

$$
\mathbb{E}\min_{\theta \in \Theta, \eta > 0} \frac{8SK^2(1+\log T)}{\eta} + \left( \frac{\hat{V}_\theta^2}{\eta} + \eta g(\theta)m + f(\theta)m + \frac{L_\eta R_\Theta}{k} + \min\left\{ \frac{\rho^2 D^2}{\eta}, \rho M \right\} \right) T
$$
$$
+ \left( \frac{4M}{\rho} + Fm \right) \sqrt{T \log k} + \frac{M(1 + \log(T+1))}{2\rho^2}
$$

(29)

*and $\sum_{t=1}^T U_t(\mathbf{x}_t, \eta_t(\theta_t), \theta_t)$ is bounded w.p. $\ge 1 - \delta 1_{k>1}$ by*

$$
\min_{\theta \in \Theta, \eta > 0} \frac{8SK^2(1+\log T)}{\eta} + \left( \frac{\hat{V}_\theta^2}{\eta} + \eta g(\theta)m + f(\theta)m + \frac{L_\eta R_\Theta}{k} + \min\left\{ \frac{\rho^2 D^2}{\eta}, \rho M \right\} \right) T
$$
$$
+ \left( \frac{4M}{\rho} + Fm \right) \left( \sqrt{T \log k} + 1_{k>1}\sqrt{\frac{T}{2}\log\frac{1}{\delta}} \right) + \frac{M(1 + \log(T+1))}{2\rho^2}
$$

(30)

*Proof.* In the following proof, we first consider online learning $U_t(\cdot, \cdot, \theta)$ for fixed $\theta \in \Theta_k$. To tune $\eta$, we online learn the one-dimensional losses $B_\theta(\mathbf{c}_\theta(\hat{\mathbf{x}}_t)\|\mathbf{c}_\theta(\mathbf{x}_t))/\eta + \eta g(\theta)$, where $\mathbf{c}_\theta(\hat{\mathbf{x}}_t)$ is the ($\eta_t(\theta)$-independent) action of FTL at time $t$. As discussed, the corresponding regularized losses $U_t^{(\rho)}$ are exp-concave, and so running EWOO yields $\tilde{\mathcal{O}}\left( M/\rho^2 + \min\left\{ \rho^2 D^2/\eta, \rho M \right\} T \right)$ regret w.r.t. the original sequence [31, Cor. C.2]. At the same time, we show that FTL has logarithmic regret on the sequence $B_\theta(\mathbf{c}_\theta(\hat{\mathbf{x}}_t)\|\cdot)$ that scales with the spectral norm $S$ of $\nabla^2 \psi_\theta$ (c.f. Lem. A.1), and that the average loss of the optimal comparator is $\hat{V}_\theta^2$ (c.f. Claim A.1). Thus, since we only care about a fixed comparator $\eta$, dividing by $\eta T$ yields the first and last terms (5). We run a copy of these algorithms for each $\theta \in \Theta_k$; since their losses are bounded by $\tilde{\mathcal{O}}(M/\rho + Fm)$, textbook results for MW yield $\mathcal{O}(\sqrt{T \log k})$ regret w.r.t. $\theta \in \Theta_k$, which we then extend to $\Theta \supset \Theta_k$ using $L_\eta$-Lipschitzness.

Formally, we have that

$$
\mathbb{E}\sum_{t=1}^T U_t(\mathbf{x}_t, \eta_t(\theta_t), \theta_t)
$$
$$
= \mathbb{E}\sum_{t=1}^T \frac{B_{\theta_t}(\mathbf{c}_{\theta_t}(\hat{\mathbf{x}}_t)\|\mathbf{x}_t)}{\eta_t(\theta_t)} + \eta_t(\theta_t)g(\theta)m + f(\theta)m
$$
$$
\le \left( M\left(\frac{1}{\rho} + \sqrt{2}\right) + Fm \right) \sqrt{2T \log k} + \mathbb{E}\min_{\theta \in \Theta_k} \sum_{t=1}^T \frac{B_\theta(\mathbf{c}_\theta(\hat{\mathbf{x}}_t)\|\mathbf{x}_t)}{\eta_t(\theta)} + \eta_t(\theta)g(\theta)m + f(\theta)m
$$
$$
\le \left( \frac{4M}{\rho} + Fm \right) \sqrt{T \log k} + \mathbb{E}\min_{\theta \in \Theta_k, \eta > 0, \mathbf{x} \in \mathcal{K}} \sum_{t=1}^T \frac{B_\theta(\mathbf{c}_\theta(\hat{\mathbf{x}}_t)\|\mathbf{x})}{\eta} + \eta g(\theta)m + f(\theta)m
$$
$$
+ \min\left\{ \frac{\rho^2 D_\theta^2}{\eta}, \rho D_\theta \sqrt{g(\theta)m} \right\} T + \frac{D_\theta \sqrt{g(\theta)m}(1 + \log(T+1))}{2\rho^2} + \frac{8SK^2(1+\log T)}{\eta}
$$

(31)

where the first inequality is the regret of multiplicative weights with step-size $\lambda$ [46, Cor. 2.14] and the second is by applying Lemma A.3 for each $\theta$. We then simplify and apply the definition of $\hat{V}_\theta^2$ via Claim A.1 and conclude by applying Lipschitzness w.r.t. $\theta$:

$$
\mathbb{E} \sum_{t=1}^{T} U_t(\mathbf{x}_t, \eta_t(\theta_t), \theta_t)
$$

$$
\leq \left( \frac{4M}{\rho} + Fm \right) \sqrt{T \log k} + \mathbb{E} \min_{\theta \in \Theta_k, \eta > 0} \frac{\hat{V}_\theta^2 T}{\eta} + \eta g(\theta) mT + f(\theta) mT
$$

$$
+ \min \left\{ \frac{\rho^2 D^2}{\eta}, \rho M \right\} T + \frac{M(1 + \log(T+1))}{2\rho^2} + \frac{8SK^2(1 + \log T)}{\eta}
$$

$$
\leq \mathbb{E} \min_{\theta \in \Theta, \eta > 0} \frac{8SK^2(1 + \log T)}{\eta} + \left( \frac{\hat{V}_\theta^2}{\eta} + \eta g(\theta) m + f(\theta) m + \frac{L_\eta R_\Theta}{k} + \min \left\{ \frac{\rho^2 D^2}{\eta}, \rho M \right\} \right) T
$$

$$
+ \left( \frac{4M}{\rho} + Fm \right) \sqrt{T \log k} + \frac{M(1 + \log(T+1))}{2\rho^2}
$$

$$
\tag{32}
$$

The w.h.p. guarantee follows by Cesa-Bianchi and Lugosi [16, Lem. 4.1]. $\qquad\square$

## B  Implicit exploration

### B.1  Properties of the Tsallis entropy

**Lemma B.1.** *For any $\varepsilon \in (0,1]$ and $\mathbf{x} \in \triangle$ s.t. $\mathbf{x}(a) \geq \frac{\varepsilon}{d} \ \forall \ a \in [d]$ the $\beta$-Tsallis entropy $H_\beta(\mathbf{x}) = -\frac{1 - \sum_{a=1}^{d} \mathbf{x}^\beta(a)}{1 - \beta}$ is $d \log \frac{d}{\varepsilon}$-Lipschitz w.r.t. $\beta \in [0,1]$.*

*Proof.* Let $\log_\beta x = \frac{x^{1-\beta} - 1}{1 - \beta}$ be the $\beta$-logarithm function and note that by Yamano [54, Equation 6] we have $\log_\beta x - \log x = (1 - \beta)(\partial_b \log_\beta x + \log_\beta x \log x) \geq 0 \ \forall \ \beta \in [0,1]$. Then we have for $\beta \in [0,1)$ that

$$
|\partial_\beta H_\beta(\mathbf{x})| = \left| \frac{-H_\beta(\mathbf{x}) - \sum_{a=1}^{d} \mathbf{x}^\beta(a) \log \mathbf{x}(a)}{1 - \beta} \right|
$$

$$
= \frac{1}{1 - \beta} \left| \sum_{a=1}^{d} \mathbf{x}^\beta(a)(\log_\beta \mathbf{x}(a) - \log \mathbf{x}(a)) \right|
$$

$$
= \frac{1}{1 - \beta} \sum_{a=1}^{d} \mathbf{x}^\beta(a)(\log_\beta \mathbf{x}(a) - \log \mathbf{x}(a)) \tag{33}
$$

$$
\leq \frac{1}{1 - \beta} \left( \sum_{a=1}^{d} \mathbf{x}(a) \right)^\beta \left( \sum_{a=1}^{d} (\log_\beta \mathbf{x}(a) - \log \mathbf{x}(a))^{\frac{1}{1-\beta}} \right)^{1-\beta}
$$

$$
\leq \frac{1}{1 - \beta} \sum_{a=1}^{d} \log_\beta \mathbf{x}(a) - \log \mathbf{x}(a) \leq \frac{d}{1 - \beta}(\log_\beta \frac{d}{\varepsilon} - \log \frac{d}{\varepsilon}) \leq -d \log \frac{d}{\varepsilon}
$$

where the fourth inequality follows by Hölder's inequality, the fifth by subadditivity of $x^a$ for $a \in (0,1]$, the sixth by the fact that $\partial_x(\log_\beta x - \log x) = x^{-\beta} - 1/x \leq 0 \ \forall \ \beta, x \in [0,1)$, and the last line by substituting $\beta = 0$ since $\partial_\beta \left( \frac{\log_\beta x - \log x}{1 - \beta} \right) = \frac{2(x - x^\beta) - (1-\beta)(x^\beta + x) \log x}{x^\beta(1-\beta)^3} \leq 0 \ \forall \ \beta \in [0,1), x \in (0, 1/d]$. For $\beta = 1$, applying L'Hôpital's rule yields

$$
\lim_{\beta \to 1} \partial_\beta H_\beta(\mathbf{x}) = -\frac{1}{2} \lim_{\beta \to 1} \sum_{a=1}^{d} \mathbf{x}^\beta(a) \log^2 \mathbf{x}(a)(1 - (1 - \beta) \log \mathbf{x}(a)) = -\frac{1}{2} \sum_{a=1}^{d} \mathbf{x}(a) \log^2 \mathbf{x}(a)
$$

$$
\tag{34}
$$

which is bounded on $[-2d/e^2, 0]$. $\qquad\square$

**Lemma B.2.** *Consider $\mathbf{x}_1, \ldots, \mathbf{x}_T \in \triangle$ s.t. $\mathbf{x}_t(a_t) = 1$ for some $a_t \in [d]$, and let $\bar{\mathbf{x}} = \frac{1}{T}\sum_{t=1}^{T}\mathbf{x}_t$ be their average. For any $\varepsilon \in (0,1]$ and $\beta \in (0,1]$ we have that for every $t \in [T]$*

$$H_\beta(\bar{\mathbf{x}}^{(\varepsilon)}) - H_\beta(\mathbf{x}_t^{(\varepsilon)}) \le H_\beta(\bar{\mathbf{x}}) \tag{35}$$

*where recall that $\mathbf{x}^{(\varepsilon)} = \mathbf{c}_{\frac{\varepsilon}{1-\varepsilon}}(\mathbf{x}) = \mathbf{1}_d/d + (1-\varepsilon)(\mathbf{x} - \mathbf{1}_d/d) = (1-\varepsilon)\mathbf{x} + \frac{\varepsilon}{d}\mathbf{1}_d$.*

*Proof.* Assume w.l.o.g. that $\bar{\mathbf{x}}(1) \le \bar{\mathbf{x}}(2) \le \ldots \le \bar{\mathbf{x}}(d)$ and $a_t = 1$, so that $\mathbf{x}_t^{(\varepsilon)} = \mathbf{e}_1^{(\varepsilon)}$. We take the derivative

$$\partial_\varepsilon H_\beta\left((1-\varepsilon)\bar{\mathbf{x}} + \frac{\varepsilon}{d}\mathbf{1}_d\right) - \partial_\varepsilon H_\beta\left(\mathbf{e}_1^{(\varepsilon)}\right)$$

$$= \frac{d}{1-\beta}\sum_{a=1}^{d-1}\left(\frac{1}{((1-\varepsilon)\bar{\mathbf{x}}(a) + \varepsilon/d)^{1-\beta}} - \frac{1}{(\varepsilon/d)^{1-\beta}}\right)$$

$$+ \frac{d}{1-\beta}\sum_{a=1}^{d-1}\left(\frac{1}{((1-\varepsilon) + \varepsilon/d)^{1-\beta}} - \frac{1}{((1-\varepsilon)\bar{\mathbf{x}}(d) + \varepsilon/d)^{1-\beta}}\right) \tag{36}$$

$$+ \frac{d^2}{1-\beta}\sum_{a=1}^{d-1}\bar{\mathbf{x}}(a)\left(\frac{1}{((1-\varepsilon)\bar{\mathbf{x}}(d) + \varepsilon/d)^{1-\beta}} - \frac{1}{((1-\varepsilon)\bar{\mathbf{x}}(a) + \varepsilon/d)^{1-\beta}}\right)$$

By the assumption that $\bar{\mathbf{x}}(a)$ is non-decreasing in $a$, each of the summands above become non-positive. So for $\varepsilon \in (0,1]$ the derivative is non-positive, and for $\varepsilon \to 0^+$ it goes to $-\infty$. Thus the l.h.s. of the bound is monotonically non-increasing in $\varepsilon$ for all $\varepsilon \in [0,1]$. The result then follows from the fact that for $\varepsilon = 0$ we have $H_\beta\left((1-\varepsilon)\bar{\mathbf{x}} + \frac{\varepsilon}{d}\mathbf{1}_d\right) - H_\beta\left(\mathbf{e}_1^{(\varepsilon)}\right) = H_\beta(\bar{\mathbf{x}})$. $\qquad\square$

## B.2 Implicit exploration bounds

**Lemma B.3.** *Suppose we play $\mathrm{OMD}_{\beta,\eta}$ with regularizer $\psi_\beta$ the negative Tsallis entropy and initialization $\mathbf{x}_1 \in \triangle$ on the sequence of linear loss functions $\ell_1, \ldots, \ell_T \in [0,1]^d$. Then for any $\mathbf{x} \in \triangle$ we have*

$$\sum_{t=1}^{T}\langle \ell_t, \mathbf{x}_t - \mathbf{x}\rangle \le \frac{B_\beta(\mathbf{x}||\mathbf{x}_1)}{\eta} + \frac{\eta}{\beta}\sum_{a=1}^{d}\mathbf{x}_t^{2-\beta}(a)\ell_t^2(a) \tag{37}$$

*Proof.* Note that the following proof follows parts of the course notes by Luo [37], which we reproduce for completeness. The OMD update at each step $t$ involves the following two steps: set $\mathbf{y}_{t+1} \in \triangle$ s.t. $\nabla\psi_\beta(\mathbf{y}_{t+1}) = \nabla\psi_\beta(\mathbf{x}_t) - \eta\ell_t$ and then set $\mathbf{x}_{t+1} = \arg\min_{\mathbf{x} \in \triangle}B_\beta(\mathbf{x}, \mathbf{y}_{t+1})$ [25, Algorithm 14]. Note that by Hazan [25, Equation 5.3] and nonnegativity of the Bregman divergence we have

$$\sum_{t=1}^{T}\langle \ell_t, \mathbf{x}_t - \mathbf{x}\rangle \le \frac{B_\beta(\mathbf{x}||\mathbf{x}_1)}{\eta} + \frac{1}{\eta}\sum_{t=1}^{T}B_\beta(\mathbf{x}_t||\mathbf{y}_{t+1}) \tag{38}$$

To bound the second term, note that when $\psi_\beta$ is the negative Tsallis entropy we have

$$B_\beta(\mathbf{x}_t||\mathbf{y}_{t+1})$$

$$= \frac{1}{1-\beta}\sum_{a=1}^{d}\left(\mathbf{y}_{t+1}^\beta(a) - \mathbf{x}_t^\beta(a) + \frac{\beta}{\mathbf{y}_{t+1}^{1-\beta}(a)}(\mathbf{x}_t(a) - \mathbf{y}_{t+1}(a))\right)$$

$$= \frac{1}{1-\beta}\sum_{a=1}^{d}\left((1-\beta)\mathbf{y}_{t+1}^\beta(a) - \mathbf{x}_t^\beta(a) + \beta\left(\frac{1}{\mathbf{x}_t^{1-\beta}(a)} + \frac{1-\beta}{\beta}\eta\ell_t(a)\right)\mathbf{x}_t(a)\right) \tag{39}$$

$$= \sum_{a=1}^{d}\left(\mathbf{y}_{t+1}^\beta(a) - \mathbf{x}_t^\beta(a) + \eta\mathbf{x}_t(a)\ell_t(a)\right)$$

Plugging the following result, which follows from $(1+x)^\alpha \leq 1 + \alpha x + \alpha(\alpha-1)x^2 \; \forall \, x \geq 0, \alpha < 0$, into the above yields the desired bound.

$$
\begin{aligned}
\mathbf{y}_{t+1}^\beta(a) = \mathbf{x}_t^\beta(a) \left( \frac{\mathbf{y}_{t+1}^{\beta-1}(a)}{\mathbf{x}_t^{\beta-1}(a)} \right)^{\frac{\beta}{\beta-1}} &= \mathbf{x}_t^\beta(a) \left( 1 + \frac{1-\beta}{\beta} \eta \mathbf{x}_t^{1-\beta}(a)\ell_t(a) \right)^{\frac{\beta}{\beta-1}} \\
&\leq \mathbf{x}_t^\beta(a) \left( 1 - \eta \mathbf{x}_t^{1-\beta}(a)\ell_t(a) + \frac{\eta^2}{\beta} \mathbf{x}_t^{2-2\beta}(a)\ell_t(a)^2 \right) \quad (40) \\
&= \mathbf{x}_t^\beta(a) - \eta \mathbf{x}_t(a)\ell_t(a) + \frac{\eta^2}{\beta} \mathbf{x}_t^{2-\beta}(a)\ell_t(a)^2
\end{aligned}
$$

$\square$

**Theorem B.1.** *In Algorithm 1, let $OMD_{\eta,\beta}$ be online mirror descent with the Tsallis entropy regularizer $\psi_\beta$ over $\gamma$-offset loss estimators, $\Theta_k$ is a subset of $[\underline{\beta}, \overline{\beta}] \subset [\frac{1}{\log d}, 1]$, and*

$$
U_t(\mathbf{x}, \eta, \beta) = \frac{B_\beta(\hat{\mathbf{x}}_t^{(\varepsilon)} \| \mathbf{x})}{\eta} + \frac{\eta d^\beta m}{\beta} \tag{41}
$$

*where $\hat{\mathbf{x}}_t^{(\varepsilon)} = (1-\varepsilon)\hat{\mathbf{x}}_t + \varepsilon \mathbf{1}_d/d$. Note that $U_t^{(\rho)}(\mathbf{x}, \eta, \beta) = U_t(\mathbf{x}, \eta, \beta) + \frac{\rho^2(d^{1-\beta}-1)}{\eta(1-\beta)}$. Then there exists settings of $\underline{\eta}, \overline{\eta}, \alpha, \lambda$ s.t. for all $\varepsilon, \rho, \gamma \in (0,1)$ we have w.p. $\geq 1 - \delta$ that*

$$
\sum_{t=1}^T \sum_{i=1}^m \ell_{t,i}(a_{t,i}) - \ell_{t,i}(\mathring{a}_t)
$$

$$
\leq (\varepsilon + \gamma d)mT + \frac{2 + \sqrt{\frac{d\log d}{em}}}{\gamma} \log \frac{5}{\delta} + \frac{8d\sqrt{m}}{\rho} \left( 1_{k>1} \sqrt{T \log \frac{5k}{\delta}} + \frac{1 + \log(T+1)}{16\rho} \right)
$$

$$
+ \min_{\beta \in [\underline{\beta}, \overline{\beta}], \eta > 0} \frac{8\left(\frac{d}{\varepsilon}\right)^{2-\underline{\beta}}(1 + \log T)}{\eta} + \left( \frac{\hat{H}_\beta}{\eta} + \frac{\eta d^\beta m}{\beta} + \frac{L_\eta(\overline{\beta} - \underline{\beta})}{2k} + d \min \left\{ \frac{\rho^2}{2\eta}, \rho\sqrt{m} \right\} \right) T
$$

$$
\tag{42}
$$

*for $L_\eta = \left( \frac{\log \frac{d}{\varepsilon}}{\eta} + \eta m \log^2 d \right) d$.*

*Proof.* In this setting we have $g(\beta) = d^\beta/\beta$, $f(\beta) = 0$, $D_\beta^2 = \frac{d^{1-\beta}-1}{1-\beta}$, $D \leq \sqrt{d/2}$, $M = d\sqrt{m}$, $F = 0$, $S = (d/\varepsilon)^{2-\underline{\beta}}$, and $K = 1$. We have that

$$
\sum_{t=1}^T \sum_{i=1}^m \ell_{t,i}(a_{t,i}) - \ell_{t,i}(\mathring{a}_t)
$$

$$
= \sum_{t=1}^T \sum_{i=1}^m \langle \hat{\ell}_{t,i}, \mathbf{x}_{t,i} \rangle - \ell_{t,i}(\mathring{a}_t) + \gamma \sum_{a=1}^d \hat{\ell}_{t,i}(a)
$$

$$
\leq \sum_{t=1}^T \frac{B_{\beta_t}(\hat{\mathbf{x}}_t^{(\varepsilon)} \| \mathbf{x}_{t,1})}{\eta_t} + \sum_{i=1}^m \langle \hat{\ell}_{t,i}, \hat{\mathbf{x}}_t^{(\varepsilon)} \rangle - \ell_{t,i}(\mathring{a}_t) + \frac{\eta_t}{\beta_t} \sum_{a=1}^d \mathbf{x}_{t,i}^{2-\beta_t}(a)\hat{\ell}_{t,i}^2(a) + \gamma \sum_{a=1}^d \hat{\ell}_{t,i}(a)
$$

$$
\leq \varepsilon mT + \sum_{t=1}^T \frac{B_{\beta_t}(\hat{\mathbf{x}}_t^{(\varepsilon)} \| \mathbf{x}_{t,1})}{\eta_t} + \sum_{i=1}^m \langle \hat{\ell}_{t,i}, \hat{\mathbf{x}}_t^{(\varepsilon)} \rangle - \langle \hat{\ell}_{t,i}, \mathring{\mathbf{x}}_t^{(\varepsilon)} \rangle
$$

$$
+ \sum_{t=1}^T \frac{\eta_t}{\beta_t} \sum_{i=1}^m \sum_{a=1}^d \mathbf{x}_{t,i}^{1-\beta_t}(a)\hat{\ell}_{t,i}(a) + \gamma \sum_{a=1}^d \hat{\ell}_{t,i}(a)
$$

$$
\tag{43}
$$

where the equality follows similarly to Luo [37] since $\langle \hat{\ell}_{t,i}, \mathbf{x}_{t,i} \rangle = \ell_{t,i}(a_{t,i}) - \gamma \sum_{a=1}^d \hat{\ell}_{t,i}(a)$, the first inequality follows by Lemma B.3 and the second by Hölder's inequality and the definitions of

$\hat{\ell}_{t,i}$ and $\hat{\mathbf{x}}_{t,i}^{(\varepsilon)}$. We next apply the optimality of $\hat{a}_t$ for $\sum_{i=1}^{m} \hat{\ell}_{t,i}$ to get

$$\sum_{t=1}^{T} \sum_{i=1}^{m} \ell_{t,i}(a_{t,i}) - \ell_{t,i}(\mathring{a}_t)$$

$$\leq \varepsilon m T + \sum_{t=1}^{T} \frac{B_{\beta_t}(\hat{\mathbf{x}}_t^{(\varepsilon)} \| \mathbf{x}_{t,1})}{\eta_t} + (1 - \varepsilon) \sum_{i=1}^{m} \hat{\ell}_{t,i}(\mathring{a}_t) - \ell_{t,i}(\mathring{a}_t) + \frac{\varepsilon}{d} \sum_{a=1}^{d} \hat{\ell}_{t,i}(a) - \ell_{t,i}(a)$$

$$+ \sum_{t=1}^{T} \frac{\eta_t}{\beta_t} \sum_{i=1}^{m} \sum_{a=1}^{d} \mathbf{x}_{t,i}^{1-\beta_t}(a) \hat{\ell}_{t,i}(a) + \gamma \sum_{a=1}^{d} \hat{\ell}_{t,i}(a)$$

$$\leq \varepsilon m T + \frac{1 + \frac{\varepsilon}{d} + \frac{\overline{\eta}}{\underline{\beta}} + \gamma}{2\gamma} \log \frac{5}{\delta} + \sum_{t=1}^{T} \frac{B_{\beta_t}(\hat{\mathbf{x}}_t^{(\varepsilon)} \| \mathbf{x}_{t,1})}{\eta_t}$$

$$+ \sum_{t=1}^{T} \frac{\eta_t}{\beta_t} \sum_{i=1}^{m} \sum_{a=1}^{d} \mathbf{x}_{t,i}^{1-\beta_t}(a) \ell_{t,i}(a) + \gamma \sum_{a=1}^{d} \ell_{t,i}(a)$$

$$\leq \varepsilon m T + \frac{2 + \sqrt{\frac{d \log d}{em}}}{\gamma} \log \frac{5}{\delta} + \gamma d m T + \sum_{t=1}^{T} \frac{B_{\beta_t}(\hat{\mathbf{x}}_t^{(\varepsilon)} \| \mathbf{x}_{t,1})}{\eta_t} + \frac{\eta_t d^{\beta_t} m}{\beta_t}$$

(44)

where the the second inequality follows by Neu [43, Lemma 1] applied to each of the last four terms and the fifth by the definition of $\ell_{t,i}$ and using $\max_{\beta \in [\frac{1}{\log d}, 1]} \overline{\eta}(\beta) \leq \sqrt{\frac{d}{em \log d}}$. Substituting into Theorem A.1 and simplifying yields the result except with $\hat{V}_{\beta}^2 = \frac{1}{T} \sum_{t=1}^{T} \psi_{\beta}(\hat{\mathbf{x}}_t^{(\varepsilon)}) - \psi_{\beta}(\hat{\bar{\mathbf{x}}}^{(\varepsilon)})$ in place of $\hat{H}_{\beta}$, but the former is bounded by the latter by Lemma B.2. $\square$

**Corollary B.1.** *Let* $\underline{\beta} = \overline{\beta} = 1$. *Then w.h.p. we can ensure task-averaged regret at most*

$$2\sqrt{\hat{H}_1 dm} + \tilde{\mathcal{O}}\left( \frac{d\sqrt{m} + d^{\frac{2}{3}} m^{\frac{2}{3}}}{\sqrt[3]{T}} \right)$$

(45)

*so long as* $mT \geq d^2$ *or alternatively ensure*

$$\min\left\{ 2\sqrt{\hat{H}_1 dm} + \tilde{\mathcal{O}}\left( \frac{d^{\frac{3}{4}} m^{\frac{3}{4}} + d\sqrt{m}}{\sqrt[4]{T}} \right), 2\sqrt{dm \log d} + \tilde{\mathcal{O}}\left( \frac{d^{\frac{3}{2}}\sqrt{m}}{\sqrt{T}} \right) \right\}$$

(46)

*so long as* $mT \geq d$.

*Proof.* Applying Theorem B.1, simplifying, and dividing by $T$ yields task-averaged regret at most

$$(\varepsilon + \gamma d)m + \frac{2 + \sqrt{\frac{d \log d}{em}}}{\gamma T} \log \frac{5}{\delta} + \left( \frac{1 + \log(T+1)}{2\rho^2 T} + \min\left\{ \frac{\rho^2}{\eta\sqrt{m}}, \rho \right\} \right) d\sqrt{m}$$

$$+ \min_{\eta > 0} \frac{8d(1 + \log T)}{\varepsilon \eta T} + \left( \frac{\hat{H}_1}{\eta} + \eta dm \right)$$

(47)

Set $\gamma = \frac{1}{\sqrt{dmT}}$. Then set $\varepsilon = \sqrt[3]{\frac{d^2}{mT}}$ and $\rho = \frac{1}{\sqrt[3]{T}}$, and use $\eta = \sqrt{\frac{\hat{H}_1}{dm}} + \frac{1}{\sqrt[3]{dmT}}$ to get the first result. Otherwise, set $\varepsilon = \sqrt{\frac{d}{mT}}$ and $\rho = \frac{1}{\sqrt[4]{T}}$, and use the better of $\eta = \sqrt{\frac{\hat{H}_1}{dm}} + \frac{1}{\sqrt[4]{dmT}}$ and $\eta = \sqrt{\frac{\log d}{dm}}$ to get the second. $\square$

**Corollary B.2.** *Let* $\underline{\beta} = \frac{1}{2}$ *and* $\overline{\beta} = 1$ *and assume* $mT \geq d^{\frac{5}{2}}$. *Then w.h.p. we can ensure task-averaged regret at most*

$$\min_{\beta \in [\frac{1}{2}, 1]} 2\sqrt{\hat{H}_{\beta} d^{\beta} m / \beta} + \tilde{\mathcal{O}}\left( \frac{d^{\frac{5}{7}} m^{\frac{5}{7}}}{T^{\frac{2}{7}}} + \frac{d\sqrt{m}}{\sqrt[4]{T}} \right)$$

(48)

*using* $k = \left\lceil \sqrt[4]{d}\sqrt{T} \right\rceil$.

*Proof.* Applying Theorem B.1, simplifying, and dividing by $T$ yields task-averaged regret at most

$$(\varepsilon + \gamma d)m + \frac{2 + \sqrt{\frac{d \log d}{em}}}{\gamma T} \log \frac{5}{\delta} + \frac{8d\sqrt{m}}{\rho} \left( \sqrt{\frac{\log \frac{5k}{\delta}}{T}} + \frac{1 + \log(T+1)}{16\rho T} \right)$$

$$+ \min_{\beta \in [\underline{\beta}, \overline{\beta}], \eta > 0} \frac{8d^{\frac{3}{2}}(1 + \log T)}{\varepsilon^{\frac{3}{2}} \eta T} + \left( \frac{\hat{H}_\beta}{\eta} + \frac{\eta d^\beta m}{\beta} + \frac{d}{4k} \left( \frac{\log \frac{d}{\varepsilon}}{\eta} + \eta m \log^2 d \right) + \rho d \sqrt{m} \right)$$

$$\tag{49}$$

Set $\gamma = \frac{1}{\sqrt{dmT}}$, $\varepsilon = \frac{d^{\frac{5}{7}}}{(mT)^{\frac{2}{7}}}$, $\rho = \frac{1}{\sqrt[4]{T}}$, and use $\eta = \sqrt{\frac{\beta \hat{H}_\beta}{md^\beta}} + \frac{1}{(dmT)^{\frac{2}{7}}}$ to get the result. $\qquad\square$

**Corollary B.3.** *Let* $\underline{\beta} = \frac{1}{\log d}$ *and* $\overline{\beta} = 1$ *and assume* $mT \geq d^3$. *Then w.h.p. we can ensure task-averaged regret at most*

$$\min_{\beta \in (0,1]} 2\sqrt{\hat{H}_\beta d^\beta m / \beta} + \tilde{\mathcal{O}} \left( \frac{d^{\frac{3}{4}} m^{\frac{3}{4}} + d\sqrt{m}}{\sqrt[4]{T}} \right) \tag{50}$$

*using* $k = \left\lceil \sqrt[4]{d}\sqrt{T} \right\rceil$.

*Proof.* Applying Theorem B.1, dividing by $T$, and simplifying yields

$$(\varepsilon + \gamma d)m + \frac{2 + \sqrt{\frac{d \log d}{em}}}{\gamma T} \log \frac{5}{\delta} + \frac{8d\sqrt{m}}{\rho} \left( \sqrt{\frac{\log \frac{5k}{\delta}}{T}} + \frac{1 + \log(T+1)}{16\rho T} \right)$$

$$+ \min_{\beta \in [\underline{\beta}, \overline{\beta}], \eta > 0} \frac{8d^2(1 + \log T)}{\varepsilon^2 \eta T} + \left( \frac{\hat{H}_\beta}{\eta} + \frac{\eta d^\beta m}{\beta} + \frac{d}{2k} \left( \frac{\log \frac{d}{\varepsilon}}{\eta} + \eta \log^2 d \right) + \rho d \sqrt{m} \right)$$

$$\tag{51}$$

Note that $\hat{H}_\beta$ and $\frac{d^\beta}{\beta}$ are both decreasing on $\beta < \frac{1}{\log d}$, so $\beta$ in the chosen interval is optimal over all $\beta \in (0, 1]$. Set $\gamma = \frac{1}{\sqrt{dmT}}$, $\varepsilon = \frac{d^{\frac{3}{4}}}{\sqrt[4]{mT}}$, $\rho = \frac{1}{\sqrt[4]{T}}$, and use $\eta = \sqrt{\frac{\beta \hat{H}_\beta}{md^\beta}} + \frac{1}{\sqrt[4]{dmT}}$ to get the result. $\quad\square$

## C Guaranteed exploration

### C.1 Best-arm identification

**Lemma C.1.** *Suppose for* $\varepsilon > 0$ *we run OMD on task* $t \in [T]$ *with initialization* $\mathbf{x}_{t,1} \in \triangle^{(\varepsilon)}$, *regularizer* $\psi_{\beta_t} + I_{\triangle^{(\varepsilon)}}$ *for some* $\beta_t \in (0, 1]$, *and unbiased loss estimators* $(\gamma = 0)$. *If Assumption 3.1 holds and* $m > \frac{28 d \log d}{3\varepsilon \Delta^2}$ *then* $\hat{\mathbf{x}}_t = \mathring{\mathbf{x}}_t$ *w.p.* $\geq 1 - d\kappa$, *where* $\kappa = \exp\left( -\frac{3\varepsilon\Delta^2 m}{28d} \right)$.

*Proof.* We extend the proof by Abbasi-Yadkori et al. [1, Appendices B and F] to arbitrary lower bounds $\varepsilon/d$ on the probability. First, since $0 \leq \hat{\ell}_{t,i}(a) \leq \frac{d}{\varepsilon} \ell_{t,i}(a)$ we have that

$$-\frac{d}{\varepsilon} \leq -1 \leq -\ell_{t,i}(a) \leq \hat{\ell}_{t,i}(a) - \ell_{t,i}(a) \leq \left( \frac{d}{\varepsilon} - 1 \right) \ell_{t,i}(a) \leq \frac{d}{\varepsilon} \tag{52}$$

and so $|\hat{\ell}_{t,i}(a) - \ell_{t,i}(a)| \leq \frac{d}{\varepsilon}$. Therefore since the variance of the estimated losses is a scaled Bernoulli we have that

$$\mathrm{Var}(\hat{\ell}_{t,i}(a) - \ell_{t,i}(a)) = \mathrm{Var}(\hat{\ell}_{t,i}(a)) = \mathbf{x}_{t,i}(a)(1 - \mathbf{x}_{t,i}(a)) \left( \frac{\ell_{t,i}(a)}{\mathbf{x}_{t,i}(a)} \right)^2 \leq \frac{\ell_{t,i}^2(a)}{\mathbf{x}_{t,i}(a)} \leq \frac{d}{\varepsilon} \tag{53}$$

We can thus apply a martingale concentration inequality of Fan et al. [23, Corollary 2.1] to the martingale difference sequence (MDS) $\frac{\varepsilon}{d}(\hat{\ell}_{t,i}(a) - \ell_{t,i}(a)) \in [-\frac{\varepsilon}{d}, 1]$ to obtain

$$
\Pr\left( \sum_{i=1}^{m} \hat{\ell}_{t,i}(a) - \ell_{t,i}(a) \geq \frac{m\Delta_a}{2} \right) = \Pr\left( \frac{\varepsilon}{d} \sum_{i=1}^{m} \hat{\ell}_{t,i}(a) - \ell_{t,i}(a) \geq \frac{\varepsilon m\Delta_a}{2d} \right)
$$

$$
\leq \Pr\left( \max_{j\in[m]} \frac{\varepsilon}{d} \sum_{i=j}^{m} \hat{\ell}_{t,i}(a) - \ell_{t,i}(a) \geq \frac{\varepsilon m\Delta_a}{2d} \right)
$$

$$
\leq \exp\left( - \frac{2\left(\frac{\varepsilon m\Delta_a}{2d}\right)^2}{\min\left\{ m(1+\varepsilon/d)^2, 4(\varepsilon m/d + \frac{\varepsilon m\Delta_a}{6})\right\}} \right) \quad (54)
$$

$$
\leq \exp\left( - \frac{2\left(\frac{\varepsilon m\Delta_a}{2d}\right)^2}{4(\varepsilon m/d + \frac{\varepsilon m\Delta_a}{6})} \right)
$$

$$
= \exp\left( - \frac{3\varepsilon m\Delta_a^2}{4d(6 + \Delta_a)} \right)
$$

$$
\leq \exp\left( - \frac{3\varepsilon m\Delta_a^2}{28d} \right)
$$

where $\Delta_a = \frac{1}{m}|\sum_{i=1}^{m} \ell_{t,i}(a) - \min_{a'\neq a} \sum_{i=1}^{m} \ell_{t,i}(a')|$ is the per-arm loss gap in the last step we apply $\Delta_a \leq 1$. For the symmetric MDS $-\frac{\varepsilon}{d} \leq \ell_{t,i}(a) - \hat{\ell}_{t,i}(a) \leq 1$ we have

$$
\Pr\left( \sum_{i=1}^{m} \hat{\ell}_{t,i}(a) - \ell_{t,i}(a) \leq -\frac{m\Delta_a}{2} \right) = \Pr\left( \sum_{i=1}^{m} \ell_{t,i}(a) - \hat{\ell}_{t,i}(a) \geq \frac{m\Delta_a}{2} \right)
$$

$$
\leq \exp\left( - \frac{2\left(\frac{m\Delta_a}{2}\right)^2}{4\left(\frac{dm}{\varepsilon} + \frac{m\Delta_a}{6}\right)} \right) \quad (55)
$$

$$
\leq \exp\left( - \frac{3\varepsilon m\Delta_a^2/d}{4(6 + \varepsilon\Delta_a/d)} \right)
$$

$$
\leq \exp\left( - \frac{3\varepsilon m\Delta_a^2}{28d} \right)
$$

We can then conclude that

$\Pr\left( \hat{\mathbf{x}}_t \neq \mathring{\mathbf{x}}_t \right)$

$$
\leq \Pr\left( \exists\, a \neq \mathring{a}_t t : \sum_{i=1}^{m} \hat{\ell}_{t,i}(a) \leq \sum_{i=1}^{m} \ell_{t,i}(\mathring{a}_t) \right)
$$

$$
\leq \Pr\left( \sum_{i=1}^{m} \hat{\ell}_{t,i}(\mathring{a}_t) \geq \sum_{i=1}^{m} \ell_{t,i}(\mathring{a}_t) + \frac{m\Delta_{\mathring{a}_t}}{2} \ \vee\ \exists\, a \neq \mathring{a}_t : \sum_{i=1}^{m} \hat{\ell}_{t,i}(a) \leq \sum_{i=1}^{m} \ell_{t,i}(a) - \frac{m\Delta_a}{2} \right)
$$

$$
\leq \Pr\left( \sum_{i=1}^{m} \hat{\ell}_{t,i}(\mathring{a}_t) \geq \sum_{i=1}^{m} \ell_{t,i}(\mathring{a}_t) + \frac{m\Delta_{\mathring{a}_t}}{2} \right) + \sum_{a\neq\mathring{a}_t} \Pr\left( \sum_{i=1}^{m} \hat{\ell}_{t,i}(a) \leq \sum_{i=1}^{m} \ell_{t,i}(a) - \frac{m\Delta_a}{2} \right)
$$

$$
\leq \exp\left( - \frac{3\varepsilon m\Delta_{\mathring{a}_t}^2}{28d} \right) + \sum_{a\neq\mathring{a}_t} \exp\left( - \frac{3\varepsilon m\Delta_a^2}{28d} \right)
$$

$$
\leq d \exp\left( - \frac{3\varepsilon m\Delta^2}{28d} \right)
$$

$$(56)$$

where the second-to-last line follows by substituting the bounds (54) and (55) into the left and right terms, respectively. □

**Lemma C.2.** *Suppose on each task $t \in [T]$ we run OMD as in Lemma C.1. Then for any $\beta \in (0,1]$ we have $\frac{1}{T}\mathbb{E}\sum_{t=1}^{T} \psi_\beta(\hat{\mathbf{x}}_t^{(\varepsilon)}) - \psi_\beta(\hat{\bar{\mathbf{x}}}^{(\varepsilon)}) \leq -\psi_\beta(\mathring{\bar{\mathbf{x}}}) + \frac{3d\kappa\beta}{1-\beta}\left( \left(\frac{d}{\varepsilon}\right)^{1-\beta} - 1 \right).$*

*Proof.* We consider the expected divergence of the best initialization under the worst-case distribution of best arm estimation, which satisfies Lemma C.1 and (56). We have by Claim A.1 and the mean-as-minimizer property of Bregman divergences that

$$
\frac{1}{T}\mathbb{E}\sum_{t=1}^{T}\psi_\beta(\hat{\mathbf{x}}_t^{(\varepsilon)}) - \psi_\beta(\hat{\bar{\mathbf{x}}}^{(\varepsilon)}) = \mathbb{E}\min_{\mathbf{x}\in\triangle^{(\varepsilon)}}\frac{1}{T}\sum_{t=1}^{T}B_\beta\left(\hat{\mathbf{x}}_t^{(\varepsilon)}||\mathbf{x}\right)
$$

$$
\leq \min_{\mathbf{x}\in\triangle^{(\varepsilon)}}\mathbb{E}\frac{1}{T}\sum_{t=1}^{T}B_\beta\left(\hat{\mathbf{x}}_t^{(\varepsilon)}||\mathbf{x}\right)
$$

$$
= \min_{\mathbf{x}\in\triangle^{(\varepsilon)}}\frac{1}{T}\sum_{t=1}^{T}\sum_{a=1}^{d}\mathbb{P}(a=\hat{a}_t)B_\beta\left(\mathbf{e}_a^{(\varepsilon)}||\mathbf{x}\right)
$$

$$
\leq \max_{\substack{\mathbf{p}_t\in\triangle,\forall t\in[T]\\ \mathbf{p}_t(a)\leq 2\kappa,\forall t\in[T],a\neq\mathring{a}_t\\ 1-d\kappa\leq\mathbf{p}_t(a),\forall t\in[T],a=\mathring{a}_t}}\min_{\mathbf{x}\in\triangle^{(\varepsilon)}}\frac{1}{T}\sum_{t=1}^{T}\sum_{a=1}^{d}\mathbf{p}_t(a)B_\beta\left(\mathbf{e}_a^{(\varepsilon)}||\mathbf{x}\right)
$$

$$(57)$$

To simplify the last expression, we define $\bar{\mathbf{p}} = \frac{1}{T}\sum_{t=1}^{T}\mathbf{p}_t$ and again apply the (weighted) mean-as-minimizer property, followed by Claim A.1:

$$
\min_{\mathbf{x}\in\triangle^{(\varepsilon)}}\frac{1}{T}\sum_{t=1}^{T}\sum_{a=1}^{d}\mathbf{p}_t(a)B_\beta\left(\mathbf{e}_a^{(\varepsilon)}||\mathbf{x}\right) = \min_{\mathbf{x}\in\triangle^{(\varepsilon)}}\sum_{a=1}^{d}\bar{\mathbf{p}}(a)B_\beta\left(\mathbf{e}_a^{(\varepsilon)}||\mathbf{x}\right) = \sum_{a=1}^{d}B_\beta\left(\mathbf{e}_a^{(\varepsilon)}||\bar{\mathbf{p}}^{(\varepsilon)}\right)
$$

$$
= \psi_\beta(\mathbf{e}_1^{(\varepsilon)}) - \psi_\beta(\bar{\mathbf{p}}^{(\varepsilon)})
$$

$$(58)$$

By substituting into the previous inequality, we can bound the expected divergence for the worst-case $\mathbf{p}_t$ as follows:

$$
\frac{1}{T}\mathbb{E}\sum_{t=1}^{T}\psi_\beta(\hat{\mathbf{x}}_t^{(\varepsilon)}) - \psi_\beta(\hat{\bar{\mathbf{x}}}^{(\varepsilon)}) \leq \psi_\beta\left(\mathbf{e}_1^{(\varepsilon)}\right) + \max_{\substack{\mathbf{p}_t\in\triangle,\forall t\in[T]\\ \mathbf{p}_t(a)\leq 2\kappa,\forall t\in[T],a\neq\mathring{a}_t\\ 1-d\kappa\leq\mathbf{p}_t(a),\forall t\in[T],a=\mathring{a}_t}}-\psi_\beta(\bar{\mathbf{p}}^{(\varepsilon)})
$$

$$
\leq \psi_\beta\left(\mathbf{e}_1^{(\varepsilon)}\right) + \max_{\substack{\sum_{t=1}^{T}\sum_{a=1}^{d}\mathbf{p}_t(a)=T\\ \sum_{t=1}^{T}\mathbf{p}_t(a)\geq(1-d\kappa)\mathring{\bar{\mathbf{x}}}(a)T,\forall a\\ \sum_{t=1}^{T}\mathbf{p}_t(a)\leq(2\kappa(1-\mathring{\bar{\mathbf{x}}}(a))T+\mathring{\bar{\mathbf{x}}}(a)T),\forall a}}-\psi_\beta(\bar{\mathbf{p}}^{(\varepsilon)})
$$

$$(59)$$

$$
= \psi_\beta\left(\mathbf{e}_1^{(\varepsilon)}\right) - \min_{\substack{\bar{\mathbf{p}}\in\triangle\\ \bar{\mathbf{p}}(a)\geq(1-d\kappa)\mathring{\bar{\mathbf{x}}}(a),\forall a\\ \bar{\mathbf{p}}(a)\leq 2\kappa+(1-2\kappa)\mathring{\bar{\mathbf{x}}}(a),\forall a}}\psi_\beta(\bar{\mathbf{p}}^{(\varepsilon)})
$$

We use the shorthand $h(\mathbf{x}) = \psi_\beta\left((1-\varepsilon)\mathbf{x} + \frac{\varepsilon}{d}\mathbf{1}_d\right)$. We have

$$
-\partial_{\mathbf{x}(a)}\left(\psi_\beta(\mathbf{x})\right) = \partial_{\mathbf{x}(a)}\left(\frac{1}{(1-\beta)}\left(\sum_{b=1}^{d}\mathbf{x}(b)^\beta - 1\right)\right)
$$

$$
= \partial_{\mathbf{x}(a)}\left(\frac{1}{(1-\beta)}\left(\sum_{b=1}^{d}\mathbf{x}(b)^\beta + \beta d^{1-\beta}(1-\sum_{b=1}^{d}\mathbf{x}(b)) - 1\right)\right) \qquad (60)
$$

$$
= \frac{\beta}{1-\beta}\cdot\left(\mathbf{x}(a)^{\beta-1} - d^{1-\beta}\right)
$$

and therefore

$$
\|\nabla h(\mathbf{x})\|_\infty = \max_{a=1,\dots,d}\left|\partial_{\mathbf{x}(a)}\psi_\beta\left((1-\varepsilon)\mathbf{x}+\frac{\varepsilon}{d}\mathbf{1}_d\right)\right|
$$

$$
\leq \frac{\beta}{1-\beta}\max_{a=1,\dots,d}\left|((1-\varepsilon)\mathbf{x}(a)+\varepsilon/d)^{\beta-1} - d^{1-\beta}\right| \qquad (61)
$$

$$
\leq \frac{\beta}{1-\beta}\left(\left(\frac{d}{\varepsilon}\right)^{1-\beta} - 1\right) = \beta\log_\beta\left(\frac{d}{\varepsilon}\right)
$$

Finally, by convexity of $h$ we have

$$
\min_{\substack{\bar{\mathbf{p}} \in \triangle \\ \bar{\mathbf{p}}(a) \geq (1-d\kappa)\overset{\circ}{\bar{\mathbf{x}}}(a), \forall a \\ \bar{\mathbf{p}}(a) \leq 2\kappa + (1-2\kappa)\overset{\circ}{\bar{\mathbf{x}}}(a), \forall a}} h(\bar{\mathbf{p}}) \geq h(\overset{\circ}{\bar{\mathbf{x}}}) - \|\nabla h(\overset{\circ}{\bar{\mathbf{x}}})\|_\infty \max_{\substack{\bar{\mathbf{p}} \in \triangle \\ \bar{\mathbf{p}}(a) \geq (1-d\kappa)\overset{\circ}{\bar{\mathbf{x}}}(a), \forall a \\ \bar{\mathbf{p}}(a) \leq 2\kappa + (1-2\kappa)\overset{\circ}{\bar{\mathbf{x}}}(a), \forall a}} \|\bar{\mathbf{p}} - \overset{\circ}{\bar{\mathbf{x}}}\|_1
$$

$$
\geq h(\overset{\circ}{\bar{\mathbf{x}}}) - 3d\kappa \|\nabla h(\overset{\circ}{\bar{\mathbf{x}}})\|_\infty \tag{62}
$$

$$
\geq h(\overset{\circ}{\bar{\mathbf{x}}}) - 3d\kappa\beta \log_\beta\left(\frac{d}{\varepsilon}\right)
$$

so we can substitute into (59) to get

$$
\frac{1}{T}\mathbb{E}\sum_{t=1}^{T} \psi_\beta(\hat{\mathbf{x}}_t^{(\varepsilon)}) - \psi_\beta(\overset{\circ}{\hat{\bar{\mathbf{x}}}}^{(\varepsilon)}) \leq -\psi_\beta(\overset{\circ}{\bar{\mathbf{x}}}^{(\varepsilon)}) + \frac{3d\kappa\beta}{1-\beta}\left(\left(\frac{d}{\varepsilon}\right)^{1-\beta} - 1\right) \tag{63}
$$

Applying Lemma B.2 completes the proof. $\qquad\square$

## C.2 Guaranteed exploration bounds

**Lemma C.3.** *Suppose we play $\mathsf{OMD}_{\beta,\eta}$ with initialization $\mathbf{x}_1 \in \triangle^{(\varepsilon)}$, regularizer $\psi_\beta + I_{\triangle^{(\varepsilon)}}$ for some $\beta \in (0,1]$, and unbiased loss estimators ($\gamma = 0$) on the sequence of loss functions $\ell_1, \ldots, \ell_T \in [0,1]^d$. Then for any $\mathring{a} \in [d]$ we have expected regret*

$$
\mathbb{E}\sum_{t=1}^{T} \ell_t(a_t) - \ell_t(\mathring{a}) \leq \frac{\mathbb{E}B_\beta(\hat{\mathbf{x}}^{(\varepsilon)}\|\mathbf{x}_1)}{\eta} + \frac{\eta d^\beta m}{\beta} + \varepsilon m \tag{64}
$$

*for $\hat{\mathbf{x}}$ the estimated optimum of the loss estimators $\hat{\ell}_1, \ldots, \hat{\ell}_T$.*

*Proof.*

$$
\begin{aligned}
\mathbb{E}\sum_{t=1}^{T} \ell_t(a_t) - \ell_t(\mathring{a}) &= \mathbb{E}\sum_{t=1}^{T} \ell_t(a_t) - \langle \ell_t, \overset{\circ}{\mathbf{x}} \rangle \\
&\leq \mathbb{E}\sum_{t=1}^{T} \ell_t(a_t) - \langle \ell_t, \overset{\circ}{\mathbf{x}}^{(\varepsilon)} \rangle + \varepsilon m \\
&= \mathbb{E}\sum_{t=1}^{m} \hat{\ell}_t(a_t) - \langle \hat{\ell}_t, \overset{\circ}{\mathbf{x}}^{(\varepsilon)} \rangle + \varepsilon m \\
&\leq \mathbb{E}\sum_{t=1}^{m} \hat{\ell}_t(a_t) - \langle \hat{\ell}_t, \hat{\mathbf{x}}^{(\varepsilon)} \rangle + \varepsilon m \\
&\leq \mathbb{E}\left(\frac{B_\beta(\hat{\mathbf{x}}^{(\varepsilon)}\|\mathbf{x}_1)}{\eta} + \frac{\eta}{\beta}\sum_{t=1}^{T}\sum_{a=1}^{d} \hat{\ell}_t^2(a)\mathbf{x}_t^{2-\beta}(a)\right) + \varepsilon m \\
&\leq \frac{\mathbb{E}B_\beta(\hat{\mathbf{x}}^{(\varepsilon)}\|\mathbf{x}_1)}{\eta} + \frac{\eta d^\beta m}{\beta} + \varepsilon m
\end{aligned} \tag{65}
$$

where the second inequality follows by optimality of $\hat{\mathbf{x}}$ for the estimated losses $\hat{\ell}_t$, the third by Lemma B.3 constrained to $\triangle^{(\varepsilon)}$, and the fourth similarly to Theorem B.1 (note both are also effectively shown in Luo [37]). $\qquad\square$

**Theorem C.1.** *In Algorithm 1, let $\mathsf{OMD}_{\eta,\beta}$ be online mirror descent with the regularizer $\psi_\beta + I_{\triangle^{(\varepsilon)}}$ over unbiased ($\gamma = 0$) loss estimators, $\Theta_k$ is a subset of $[\underline{\beta}, \overline{\beta}] \subset [\frac{1}{\log d}, 1]$, and*

$$
U_t(\mathbf{x}, \eta, \beta) = \frac{B_\beta(\hat{\mathbf{x}}_t^{(\varepsilon)}\|\mathbf{x})}{\eta} + \frac{\eta d^\beta m}{\beta} \tag{66}
$$

where $\hat{\mathbf{x}}_t^{(\varepsilon)} = (1-\varepsilon)\hat{\mathbf{x}}_t + \varepsilon\mathbf{1}_d/d$. Note that $U_t^{(\rho)}(\mathbf{x}, \eta, \beta) = U_t(\mathbf{x}, \eta, \beta) + \frac{\rho^2(d^{1-\beta}-1)}{\eta(1-\beta)}$. Then under Assumption 3.1 there exists settings of $\underline{\eta}, \overline{\eta}, \alpha, \lambda$ s.t. for all $\varepsilon, \rho \in (0,1)$ we have that

$$\mathbb{E}\frac{1}{T}\sum_{t=1}^{T}\sum_{i=1}^{m}\ell_{t,i}(a_{t,i}) - \ell_{t,i}(\mathring{a}_t)$$

$$\leq \varepsilon m + \frac{8d\sqrt{m}}{\rho}\left(1_{k>1}\sqrt{\frac{\log k}{T}} + \frac{1+\log(T+1)}{16\rho T}\right)$$

$$+ \min_{\beta\in[\underline{\beta},\overline{\beta}],\eta>0}\frac{8\left(\frac{d}{\varepsilon}\right)^{2-\beta}(1+\log T)}{\eta T} + \frac{h_\beta(\Delta)}{\eta} + \frac{\eta d^\beta m}{\beta} + \frac{L_\eta(\overline{\beta}-\underline{\beta})}{2k} + d\min\left\{\frac{\rho^2}{2\eta}, \rho\sqrt{m}\right\} \tag{67}$$

for $L_\eta = \left(\frac{\log\frac{d}{\varepsilon}}{\eta} + \eta m \log^2 d\right)d$ and $h_\beta(\Delta) = (H_\beta + \frac{56}{dm})\iota_\Delta + \frac{d^{1-\beta}-1}{1-\beta}(1-\iota_\Delta)$ for $\iota_\Delta = 1_{m\geq\frac{75d}{\varepsilon\Delta^2}\log\frac{d}{\varepsilon\Delta^2}}$.

*Proof.* By Lemma C.3 we have

$$\mathbb{E}\sum_{t=1}^{T}\sum_{i=1}^{m}\ell_{t,i}(a_{t,i}) - \ell_{t,i}(\mathring{a}_t) \leq \varepsilon mT + \mathbb{E}\sum_{t=1}^{T}\frac{B_{\beta_t}(\hat{\mathbf{x}}_t^{(\varepsilon)}\|\mathbf{x}_{t,1})}{\eta_t} + \frac{\eta_t d^{\beta_t}m}{\beta_t} \tag{68}$$

Since we have the same environment-dependent quantities as in Theorem B.1, we can substitute the above bound into Theorem A.1 and then apply the Lemma C.2 bound

$$\mathbb{E}\hat{V}_\beta^2 \leq H_\beta + \frac{3d\kappa\beta}{1-\beta}\left(\left(\frac{d}{\varepsilon}\right)^{1-\beta} - 1\right) \leq H_\beta + \frac{3d^2}{\varepsilon}\exp\left(-\frac{3\varepsilon\Delta^2 m}{28d}\right)$$

$$= H_\beta + \frac{3\varepsilon\Delta^2}{d^2}\exp\left(4\log\frac{d}{\varepsilon\Delta^2} - \frac{3\varepsilon\Delta^2 m}{28d}\right) \tag{69}$$

$$\leq H_\beta + \frac{3\varepsilon\Delta^2/d^2}{\frac{3\varepsilon\Delta^2 m}{28d} - 4\log\frac{d}{\varepsilon\Delta^2}}$$

$$\leq H_\beta + \frac{56}{dm}$$

where the last line follows by assuming $m \geq \frac{75d}{\varepsilon\Delta^2}\log\frac{d}{\varepsilon\Delta^2}$. If this condition does not hold, then we apply the default bound of $\mathbb{E}\hat{V}_\beta^2 \leq= \frac{1}{T}\sum_{t=1}^{T}\psi_\beta(\hat{\mathbf{x}}_t) - \psi_\beta(\hat{\mathbf{x}}) \leq \frac{d^{1-\beta}-1}{1-\beta}$. $\square$

**Corollary C.1.** *Let $\underline{\beta} = \overline{\beta} = 1$. Then for known $\Delta$ and assuming $m \geq \frac{75d}{\Delta^2}\log\frac{d}{\Delta^2}$ we can ensure expected task-averaged regret at most*

$$2\sqrt{H_1 dm + 56} + \frac{75d}{\Delta^2}W\left(\frac{m}{75}\right) + \tilde{\mathcal{O}}\left(\frac{d^{\frac{3}{2}}m^{\frac{3}{4}}}{\sqrt{T}} + \frac{d\Delta^2 m^2}{T}\right) \tag{70}$$

*where $W$ is the Lambert $W$-function, while for unknown $\Delta$ we can ensure expected task-averaged regret at most*

$$2\sqrt{H_1 dm + 56} + \frac{3}{\Delta}\sqrt[3]{50dm\log d\log\frac{d^2 m^2}{150\Delta^6\log d}} + \tilde{\mathcal{O}}\left(\frac{d^{\frac{3}{2}}m^{\frac{3}{4}}}{\sqrt{T}} + \frac{d^{\frac{4}{3}}m^{\frac{5}{3}}}{T}\right) \tag{71}$$

*so long as $m^2 \geq 150d\log d$.*

*Proof.* Applying Theorem C.1 and simplifying yields

$$\varepsilon m + \frac{8d\sqrt{m}(1+\log(T+1))}{16\rho^2 T} + \min_{\eta>0}\frac{8d(1+\log T)}{\varepsilon\eta T} + \frac{h_1(\Delta)}{\eta} + \eta dm + \frac{d\rho^2}{2\eta} \tag{72}$$

Then substitute $\eta = \sqrt{\frac{h_1(\Delta)}{dm}}$ and set $\rho = \sqrt[4]{\frac{1}{dT\sqrt{m}}}$ and $\varepsilon = \frac{75d}{\Delta^2 m}W(\frac{m}{75})$ (for known $\Delta$) or $\varepsilon = \sqrt[3]{\frac{150d\log d}{m^2}}$ (otherwise). $\square$

**Corollary C.2.** *Let $\underline{\beta} = \frac{1}{2}$ and $\overline{\beta} = 1$. Then for known $\Delta$ and assuming $m \geq \frac{75d}{\Delta^2} \log \frac{d}{\Delta^2}$ we can ensure task-averaged regret at most*

$$\min_{\beta \in [\frac{1}{2}, 1]} 2\sqrt{(H_\beta m + 56/d)d^\beta/\beta} + \frac{75d}{\Delta^2} W\left(\frac{m}{75}\right) + \tilde{\mathcal{O}}\left(\frac{d^{\frac{4}{3}} m^{\frac{2}{3}}}{\sqrt[3]{T}} + \frac{d^{\frac{5}{3}} m^{\frac{5}{6}}}{T^{\frac{2}{3}}} + \frac{d\Delta^3 m^{\frac{5}{2}}}{T}\right) \quad (73)$$

*using $k = \lceil \sqrt[3]{d^2 m T} \rceil$, while for unknown $\Delta$ we can ensure expected task-averaged regret at most*

$$\min_{\beta \in [\frac{1}{2}, 1]} 2\sqrt{(H_\beta m + 56/d)d^\beta/\beta} + \frac{3}{\Delta} \sqrt[3]{50d^2 m \log \frac{dm^2}{150\Delta^6}} + \tilde{\mathcal{O}}\left(\frac{d^{\frac{4}{3}} m^{\frac{2}{3}}}{\sqrt[3]{T}} + \frac{d^{\frac{5}{3}} m^{\frac{5}{6}}}{T^{\frac{2}{3}}} + \frac{d^{\frac{3}{2}} m^2}{T}\right)$$
$$(74)$$

*so long as $m \geq 5d\sqrt{6}$.*

*Proof.* Applying Theorem C.1 and simplifying yields

$$\varepsilon m + \frac{8d\sqrt{m}}{\rho}\left(\sqrt{\frac{\log k}{T}} + \frac{1 + \log(T+1)}{16\rho T}\right)$$
$$(75)$$
$$+ \min_{\beta \in [\underline{\beta}, \overline{\beta}], \eta > 0} \frac{8d^{\frac{3}{2}}(1 + \log T)}{\varepsilon^{\frac{3}{2}} \eta T} + \frac{h_\beta(\Delta)}{\eta} + \frac{\eta d^\beta m}{\beta} + \frac{d}{4k}\left(\frac{\log \frac{d}{\varepsilon}}{\eta} + \eta m \log^2 d\right) + \frac{d\rho^2}{2\eta}$$

Then substitute $\eta = \sqrt{\frac{h_\beta(\Delta)}{d^\beta m/\beta}}$ and set $\rho = \sqrt[3]{\frac{1}{d\sqrt{mT}}}$ and $\varepsilon = \frac{75d}{\Delta^2 m} W(\frac{m}{75})$ (for known $\Delta$) or $\varepsilon = \sqrt[3]{\frac{150d^2}{m^2}}$ (otherwise). $\qquad\square$

**Corollary C.3.** *Let $\underline{\beta} = \frac{1}{\log d}$ and $\overline{\beta} = 1$. Then for known $\Delta$ and assuming $m \geq \frac{75d}{\Delta^2} \log \frac{d}{\Delta^2}$ we can ensure task-averaged regret at most*

$$\min_{\beta \in (0, 1]} 2\sqrt{(H_\beta m + 56/d)d^\beta/\beta} + \frac{75d}{\Delta^2} W\left(\frac{m}{75}\right) + \tilde{\mathcal{O}}\left(\frac{d^{\frac{4}{3}} m^{\frac{2}{3}}}{\sqrt[3]{T}} + \frac{d^{\frac{5}{3}} m^{\frac{5}{6}}}{T^{\frac{2}{3}}} + \frac{d\Delta^4 m^3}{T}\right) \quad (76)$$

*using $k = \lceil \sqrt[3]{d^2 m T} \rceil$, while for unknown $\Delta$ we can ensure expected task-averaged regret at most*

$$\min_{\beta \in (0, 1]} 2\sqrt{(H_\beta m + 56/d)d^\beta/\beta} + \frac{3}{\Delta} \sqrt[3]{50d^2 m \log \frac{dm^2}{150\Delta^6}} + \tilde{\mathcal{O}}\left(\frac{d^{\frac{4}{3}} m^{\frac{2}{3}}}{\sqrt[3]{T}} + \frac{d^{\frac{5}{3}} m^{\frac{5}{6}}}{T^{\frac{2}{3}}} + \frac{d^{\frac{5}{3}} m^{\frac{7}{3}}}{T}\right)$$
$$(77)$$

*so long as $m \geq 5d\sqrt{6}$.*

*Proof.* Applying Theorem C.1 and simplifying yields

$$\varepsilon m + \frac{8d\sqrt{m}}{\rho}\left(\sqrt{\frac{\log k}{T}} + \frac{1 + \log(T+1)}{16\rho T}\right)$$
$$(78)$$
$$+ \min_{\beta \in [\underline{\beta}, \overline{\beta}], \eta > 0} \frac{8d^2(1 + \log T)}{\varepsilon^2 \eta T} + \frac{h_\beta(\Delta)}{\eta} + \frac{\eta d^\beta m}{\beta} + \frac{d}{2k}\left(\frac{\log \frac{d}{\varepsilon}}{\eta} + \eta m \log^2 d\right) + \frac{d\rho^2}{2\eta}$$

Then substitute $\eta = \sqrt{\frac{h_\beta(\Delta)}{d^\beta m/\beta}}$ and set $\rho = \sqrt[3]{\frac{1}{d\sqrt{mT}}}$ and $\varepsilon = \frac{75d}{\Delta^2 m} W(\frac{m}{75})$ (for known $\Delta$) or $\varepsilon = \sqrt[3]{\frac{150d^2}{m^2}}$ (otherwise). $\qquad\square$

**Corollary C.4.** *Let $\underline{\beta} = \frac{1}{\log d}$ and $\overline{\beta} = 1$. Then for unknown $\Delta$ and assuming $m \geq \max\{d^{\frac{3}{4}}, 56\}$ we can ensure task-averaged regret at most*

$$\min_{\beta \in (0, 1]} \min\left\{8\sqrt{dm}, 2\sqrt{\left(H_\beta m + \frac{56}{d}\right)\frac{d^\beta}{\beta}} + \frac{21d^{\frac{3}{4}}\sqrt[3]{m}}{\Delta}\sqrt{3\log \frac{dm}{\Delta^2}}\right\} + \tilde{\mathcal{O}}\left(\frac{d^{\frac{4}{3}} m^{\frac{2}{3}}}{\sqrt[3]{T}} + \frac{d^{\frac{5}{3}} m^{\frac{5}{6}}}{T^{\frac{2}{3}}} + \frac{d^2 m^{\frac{7}{3}}}{T}\right)$$
$$(79)$$

*using $k = \lceil \sqrt[3]{d^2 m T} \rceil$.*

*Proof.* Applying Theorem C.1 and simplifying yields

$$
\varepsilon m + \frac{8d\sqrt{m}}{\rho}\left(\sqrt{\frac{\log k}{T}} + \frac{1 + \log(T+1)}{16\rho T}\right)
$$

$$
+ \min_{\beta \in [\underline{\beta},\overline{\beta}], \eta > 0} \frac{8d^2(1 + \log T)}{\varepsilon^2 \eta T} + \frac{h_\beta(\Delta)}{\eta} + \frac{\eta d^\beta m}{\beta} + \frac{d}{2k}\left(\frac{\log\frac{d}{\varepsilon}}{\eta} + \eta m \log^2 d\right) + \frac{d\rho^2}{2\eta} \tag{80}
$$

Then substitute $\eta = \sqrt{\frac{h_\beta(\Delta)}{d^\beta m / \beta}}$ and set $\rho = \sqrt[3]{\frac{1}{d\sqrt{mT}}}$ and $\varepsilon = \frac{\sqrt{d}}{\sqrt[3]{m^2}}$. $\qquad\square$

## D   Robustness to outliers

**Proposition D.1.** *Suppose there exists a constant $p \in [0,1]$ and a subset $S \subset [T]$ of size $s$ such that $\mathring{a}_t \in S$ for all but $\mathcal{O}(T^p)$ MAB tasks $t \in [T]$. Then if $\beta \in [\frac{1}{\log d}, \frac{1}{2}]$ we have $H_\beta = \mathcal{O}(s + \frac{d^{1-\beta}}{T^{\beta(1-p)}})$.*

*Proof.* Define the vector $\mathbf{e}_S \in [0,1]^d$ s.t. $\mathbf{e}_{S[a]} = 1_{a \in S}$. Then by Claim A.1 and the mean-as-minimizer property of Bregman divergences we have

$$
H_\beta = -\psi_\beta(\mathring{\bar{\mathbf{x}}})
$$

$$
= \frac{1}{T}\sum_{t=1}^T \psi_\beta(\mathring{\mathbf{x}}_t) - \psi_\beta(\mathring{\bar{\mathbf{x}}})
$$

$$
= \frac{1}{T}\sum_{t=1}^T B_\beta(\mathring{\mathbf{x}}_t \| \mathring{\bar{\mathbf{x}}})
$$

$$
= \min_{\mathbf{x} \in \triangle_d} \frac{1}{T}\sum_{t=1}^T B_\beta(\mathring{\mathbf{x}}_t \| \mathring{\bar{\mathbf{x}}})
$$

$$
\le \min_{\delta \in (0,1)} \frac{1}{T}\sum_{t=1}^T B_\beta\left(\mathring{\mathbf{x}}_t \left\| \frac{1-\delta}{s}\mathbf{e}_S + \frac{\delta}{d}\mathbf{1}_d\right.\right)
$$

$$
= \min_{\delta \in (0,1)} \frac{1}{T}\sum_{t=1}^T \frac{1}{1-\beta}\sum_{a=1}^d \left(\frac{1-\delta}{s}1_{a\in S} + \frac{\delta}{d}\right)^\beta - \mathring{\mathbf{x}}_{t[a]}^\beta + \frac{\beta\left(\mathring{\mathbf{x}}_{t[a]} - \frac{1-\delta}{s}1_{a\in S} - \frac{\delta}{d}\right)}{\left(\frac{1-\delta}{s}1_{s\in S} + \frac{\delta}{d}\right)^\beta} \tag{81}
$$

$$
= \min_{\delta \in (0,1)} \frac{1}{T}\sum_{t=1}^T\sum_{a=1}^d \left(\frac{1-\delta}{s}1_{a\in S} + \frac{\delta}{d}\right)^\beta - \frac{\mathring{\mathbf{x}}_{t[a]}^\beta}{1-\beta} + \frac{\beta\mathring{\mathbf{x}}_{t[a]}^\beta}{(1-\beta)\left(\frac{1-\delta}{s}1_{a\in S} + \frac{\delta}{d}\right)^{1-\beta}}
$$

$$
\le \min_{\delta \in (0,1)} s^{1-\beta} + \delta^\beta d^{1-\beta} + \frac{\beta}{(1-\beta)T}\sum_{t=1}^T\sum_{a=1}^d \frac{1_{a=\mathring{a}_t}}{(1-\beta)\left(\frac{1-\delta}{s}1_{a\in S} + \frac{\delta}{d}\right)^{1-\beta}}
$$

$$
\le \min_{\delta \in (0,1)} \frac{s^{1-\beta}}{1-\beta} + \delta^\beta d^{1-\beta} + \mathcal{O}\left(\frac{\beta\left(\frac{d}{\delta}\right)^{1-\beta}}{(1-\beta)T^{1-p}}\right)
$$

$$
= \mathcal{O}\left(s + \frac{d^{1-\beta}}{T^{\beta(1-p)}}\right)
$$

where the last line follows by considering $\delta = 1/T^{1-p}$. $\qquad\square$

# E    Online learning with self-concordant barrier regularizers

## E.1    General results

**Lemma E.1.** *Let $\mathcal{K} \subset \mathbb{R}^d$ be a convex set and $\psi : \mathcal{K}^\circ \mapsto \mathbb{R}^d$ be a self-concordant barrier. Suppose $\ell_1, \ldots, \ell_T$ are a sequence of loss functions satisfying $|\langle \ell_t, \mathbf{x} \rangle| \leq 1 \; \forall \; \mathbf{x} \in \mathcal{K}$. Then if we run OMD with step-size $\eta > 0$ as in Abernethy et al. [2, Alg. 1] on the sequence of estimators $\hat{\ell}_t$ our estimated regret w.r.t. any $\mathbf{x} \in \mathcal{K}_\varepsilon$ for $\varepsilon > 0$ will satisfy*

$$\sum_{t=1}^{T} \langle \hat{\ell}_t, \mathbf{x}_t - \mathbf{x} \rangle \leq \frac{B(\mathbf{x} \| \mathbf{x}_1)}{\eta} + 32 d^2 \eta T \tag{82}$$

*Proof.* The result follows from Abernethy et al. [2] by stopping the derivation on the second inequality below Equation 10. $\qquad\square$

**Definition E.1.** *For any convex set $\mathcal{K}$ and any point $\mathbf{y} \in \mathcal{K}$, $\pi_{\mathbf{y}}(\mathbf{x}) = \inf\limits_{t \geq 0, \mathbf{y} + \frac{\mathbf{x}-\mathbf{y}}{t} \in \mathcal{K}} t$ is the* **Minkowski function with pole $\mathbf{y}$**.

**Lemma E.2.** *For any $\mathbf{x} \in \mathcal{K} \subset \mathbb{R}^d$ and $\psi : \mathcal{K}^\circ \mapsto \mathbb{R}$ a $\nu$-self-concordant regularizer with minimum $\mathbf{x}_1 \in \mathcal{K}^\circ$, the quantity $\psi(\mathbf{c}_\varepsilon(\mathbf{x}))$ is $\nu\sqrt{2}$-Lipschitz w.r.t. $\varepsilon \in [0, 1]$.*

*Proof.* Consider any $\varepsilon, \varepsilon' \in [0, 1]$ s.t. $\varepsilon' - \varepsilon \in (0, \frac{1}{2}]$ Note that for $t = \frac{\varepsilon' - \varepsilon}{1 + \varepsilon}$ we have

$$\mathbf{c}_{\varepsilon'}(\mathbf{x}) + \frac{\mathbf{c}_{\varepsilon'}(\mathbf{x}) - \mathbf{c}_\varepsilon(\mathbf{x})}{t} = \mathbf{x}_1 + \frac{\mathbf{x} - \mathbf{x}_1}{1 + \varepsilon'} + \frac{\mathbf{x}_1 + \frac{\mathbf{x}-\mathbf{x}_1}{1+\varepsilon} - \mathbf{x}_1 - \frac{\mathbf{x}-\mathbf{x}_1}{1+\varepsilon'}}{t} = \mathbf{x} \in \mathcal{K} \tag{83}$$

so $\pi_{\mathbf{c}_{\varepsilon'}(\mathbf{x})}(\mathbf{c}_\varepsilon(\mathbf{x})) \leq \frac{\varepsilon' - \varepsilon}{1 + \varepsilon} \leq \varepsilon' - \varepsilon$. Therefore by Nesterov and Nemirovskii [42, Prop. 2.3.2] we have

$$\psi(\mathbf{c}_\varepsilon(\mathbf{x})) - \psi(\mathbf{c}_{\varepsilon'}(\mathbf{x})) \leq \nu \log \left( \frac{1}{1 - \pi_{\mathbf{c}_{\varepsilon'}(\mathbf{x})}(\mathbf{c}_\varepsilon(\mathbf{x}))} \right) \leq \nu \log \left( \frac{1}{1 + \varepsilon - \varepsilon'} \right) \leq \nu(\varepsilon' - \varepsilon)\sqrt{2} \tag{84}$$

where for the last inequality we used $-\log(1 - x) \leq x\sqrt{2}$ for $x \in [0, \frac{1}{2}]$. The case of $\varepsilon' - \varepsilon \in (0, 1]$ follows by considering $\varepsilon'' = \frac{\varepsilon' + \varepsilon}{2}$ and applying the above twice. $\qquad\square$

**Theorem E.1.** *In Algorithm 1, let* $\mathrm{OMD}_{\eta,\varepsilon}$ *be online mirror descent over loss estimators specified in Abernethy et al. [2] with a $\nu$-self-concordant barrier regularizer $\psi : \mathcal{K}^\circ \mapsto \mathbb{R}$ that satisfies $\nu \geq 1$ and $\|\nabla^2 \psi(\mathbf{x}_1)\|_2 = S_1 \geq 1$. Let $\Theta_k$ be a subset of $[\frac{1}{m}, 1]$ and*

$$U_t(\mathbf{x}, \eta, \varepsilon) = \frac{B(\mathbf{c}_\varepsilon(\hat{\mathbf{x}})\|\mathbf{x})}{\eta} + 32\eta d^2 + \varepsilon m \tag{85}$$

*Note that $U_t^{(\rho)}(\mathbf{x}, \eta, \varepsilon) = U_t(\mathbf{x}, \eta, \varepsilon) + \frac{9\nu^{\frac{3}{2}}\rho^2 Km\sqrt{S_1}}{\eta}$. Then there exists settings of $\underline{\eta}, \overline{\eta}, \alpha, \lambda$ s.t. for all $\varepsilon, \rho \in (0, 1)$ we have expected task averaged regret at most*

$$\mathbb{E} \min_{\varepsilon \in [\frac{1}{m}, 1], \eta > 0} \frac{512\nu^2 K^2 S_1 m^2 (1 + \log T)}{\eta} + \left( \frac{\hat{V}_\varepsilon^2}{\eta} + 32\eta d^2 m + \varepsilon m + \frac{\nu\sqrt{2}/\eta + m}{k} \right) T$$

$$+ 3\nu^{\frac{3}{4}} m \min \left\{ \frac{3\rho^2 \nu^{\frac{3}{4}} K\sqrt{S_1}}{\eta}, 4d\rho\sqrt{2K\sqrt{S_1}} \right\} T \tag{86}$$

$$+ \frac{7dm}{\rho} \sqrt{2K\sqrt{\nu^3 S_1}} \left( 7\sqrt{T \log k} + \frac{1 + \log(T+1)}{\rho} \right)$$

*Proof.* Let $\underline{\varepsilon} = \frac{1}{m}$. For any $\varepsilon \in [\underline{\varepsilon}, 1]$ and $\mathbf{x} \in \mathcal{K}$ we have $\pi_{\mathbf{x}_1}(\mathbf{c}_\varepsilon(\mathbf{x})) \leq \frac{1}{1+\varepsilon}$, so by Nesterov and Nemirovskii [42, Prop. 2.3.2] we have

$$\|\nabla^2 \psi(\mathbf{c}_\varepsilon(\mathbf{x}))\|_2 \leq \left( \frac{1 + 3\nu}{1 - \pi_{\mathbf{x}_1}(\mathbf{c}_\varepsilon(\mathbf{x}))} \right)^2 \|\nabla^2 \psi(\mathbf{x}_1)\|_2 \leq \frac{64\nu^2 S_1}{\varepsilon^2} \tag{87}$$

Thus $S = \max_{\mathbf{x}, \mathbf{y} \in \mathcal{K}, \varepsilon \in [\underline{\varepsilon}, 1]} \|\nabla^2 \psi(\mathbf{c}_\varepsilon(\mathbf{x}))\|_2 = \frac{64\nu^2 S_1}{\underline{\varepsilon}^2}$ and also

$$\begin{aligned}
D_\varepsilon^2 &= \max_{\mathbf{x}, \mathbf{y} \in \mathcal{K}} B(\mathbf{c}_\varepsilon(\mathbf{x})\|\mathbf{c}_\varepsilon(\mathbf{y})) \\
&= \max_{\mathbf{x}, \mathbf{y} \in \mathcal{K}} \psi(\mathbf{c}_\varepsilon(\mathbf{x})) - \psi(\mathbf{c}_\varepsilon(\mathbf{y})) - \langle \nabla \psi(\mathbf{c}_\varepsilon(\mathbf{y})), \mathbf{x} - \mathbf{y} \rangle \\
&\leq \max_{\mathbf{x}, \mathbf{y} \in \mathcal{K}} \nu \log \left( \frac{1}{1 - \pi_{\mathbf{x}_1}(\mathbf{c}_\varepsilon(\mathbf{x}))} \right) + \sqrt{\nu \|\nabla^2 \psi(\mathbf{c}_\varepsilon(\mathbf{y}))\|_2} \|\mathbf{x} - \mathbf{y}\|_2 \\
&\leq \nu \log \frac{2}{\varepsilon} + \frac{8\nu^{\frac{3}{2}} K\sqrt{S_1}}{\varepsilon} \\
&\leq \frac{9\nu^{\frac{3}{2}} K\sqrt{S_1}}{\varepsilon}
\end{aligned} \tag{88}$$

where the first inequality follows by Nesterov and Nemirovskii [42, Prop. 2.3.2] and the definition of a self-concordant barrier [2, Def. 5]. In addition, we have $g(\varepsilon) = 32d^2$, $f(\varepsilon) = \varepsilon$, $M = 12d\sqrt{2Km/\underline{\varepsilon}}\sqrt[4]{\nu^3 S_1}$, and $F = 1$. We have

$$\begin{aligned}
\mathbb{E} \sum_{t=1}^{T} \sum_{i=1}^{m} \langle \ell_{t,i}, \mathbf{x}_{t,i} - \mathring{\mathbf{x}}_t \rangle &\leq \mathbb{E} \sum_{t=1}^{T} \varepsilon_t m + \sum_{i=1}^{m} \langle \ell_{t,i}, \mathbf{x}_{t,i} - \mathbf{c}_{\varepsilon_t}(\mathring{\mathbf{x}}_t) \rangle \\
&\leq \mathbb{E} \sum_{t=1}^{T} \varepsilon_t m + \sum_{i=1}^{m} \langle \hat{\ell}_{t,i}, \mathbf{x}_{t,i} - \mathbf{c}_{\varepsilon_t}(\mathring{\mathbf{x}}_t) \rangle \\
&\leq \mathbb{E} \sum_{t=1}^{T} \varepsilon_t m + \sum_{i=1}^{m} \langle \hat{\ell}_{t,i}, \mathbf{x}_{t,i} - \mathbf{c}_{\varepsilon_t}(\hat{\mathbf{x}}_t) \rangle \\
&\leq \sum_{t=1}^{T} \frac{\mathbb{E} B(\mathbf{c}_{\varepsilon_t}(\hat{\mathbf{x}}_t)\|\mathbf{x}_{t,1})}{\eta_t} + (32\eta_t d^2 + \varepsilon_t) m
\end{aligned} \tag{89}$$

where the first inequality follows by Abernethy et al. [2, Lem. 8], the second by Abernethy et al. [2, Lem. 3], the third by optimality of $\hat{\mathbf{x}}_t$, and the fourth by Lemma E.1. Substituting into Theorem A.1 and simplifying yields the result. $\qquad\square$

## E.2 Specialization to the unit sphere

**Corollary E.1.** *Let $\mathcal{K}$ be the unit sphere with the self-concordant barrier $\psi(\mathbf{x}) = -\log(1 - \|\mathbf{x}\|_2^2)$. Then Algorithm 1 attains expected task-averaged regret bounded by*

$$\tilde{\mathcal{O}}\left(\frac{dm^{\frac{3}{2}}}{T^{\frac{3}{4}}} + \frac{dm}{\sqrt[4]{T}}\right) + \min_{\varepsilon \in [\frac{1}{m}, 1]} 4d\sqrt{2m\log\left(1 + \frac{1 - \mathbb{E}\|\hat{\hat{\mathbf{x}}}\|_2^2}{2\varepsilon + \varepsilon^2}\right)} + \varepsilon m \tag{90}$$

*using $k = \lceil \sqrt{T} \rceil$.*

*Proof.* Using the fact the $\nu = 1$ and $K = S_1 = 2$, we apply Theorem E.1 and simplify to obtain

$$\mathbb{E}\min_{\varepsilon \in [\frac{1}{m}, 1], \eta > 0} \frac{\hat{V}_\varepsilon^2}{\eta} + 32\eta d^2 m + \varepsilon m + \tilde{\mathcal{O}}\left(\frac{m^2}{\eta T} + \frac{1}{\eta k} + \frac{m}{k} + m\min\left\{\frac{\rho^2}{\eta}, d\rho\right\} + \frac{dm}{\rho\sqrt{T}} + \frac{dm}{\rho^2 T}\right) \tag{91}$$

Then substitute $\eta = \frac{\hat{V}_\varepsilon}{4\sqrt{2dm}} + \frac{\sqrt{m}}{d\sqrt[4]{T}}$, set $\rho = \frac{1}{\sqrt[4]{T}}$, and note that

$$\mathbb{E}\hat{V}_\varepsilon = \mathbb{E}\sqrt{\log\left(\frac{1 - \|\mathbf{c}_\varepsilon(\hat{\hat{\mathbf{x}}})\|_2^2}{\sqrt[T]{\prod_{t=1}^T 1 - \|\mathbf{c}_\varepsilon(\hat{\mathbf{x}}_t)\|_2^2}}\right)} = \mathbb{E}\sqrt{\log\left(\frac{1 - (1+\varepsilon)^{-2}\|\hat{\hat{\mathbf{x}}}\|_2^2}{1 - (1+\varepsilon)^{-2}}\right)} \tag{92}$$

$$\leq \sqrt{\log\left(1 + \frac{1 - \mathbb{E}\|\hat{\hat{\mathbf{x}}}\|_2^2}{2\varepsilon + \varepsilon^2}\right)}$$

where we use the fact that $\|\hat{\mathbf{x}}_t\|_2 = 1$ and the inequality is Jensen's. $\square$

## E.3 Specialization to polytopes, specifically the bandit online shortest-path problem

**Corollary E.2.** *Let $\mathcal{K} = \{\mathbf{x} \in [0,1]^{|E|} : \langle \mathbf{a}, \mathbf{x} \rangle \leq b \ \forall \ (\mathbf{a}, b) \in \mathcal{C}\}$ be the set of flows from $u$ to $v$ on a graph $G(V, E)$, where $\mathcal{C} \subset \mathbb{R}^{|E|} \times \mathbb{R}$ is a set of $\mathcal{O}(|E|)$ linear constraints. Suppose we see $T$ instances of the bandit online shortest path problem with $m$ timesteps each. Then sampling from probability distributions over paths from $u$ to $v$ returned by running Algorithm 1 with regularizer $\psi(\mathbf{x}) = -\sum_{\mathbf{a},b \in \mathcal{C}} \log(b - \langle \mathbf{a}, \mathbf{x} \rangle)$ attains the following expected average regret across instances*

$$\tilde{\mathcal{O}}\left(\frac{|E|^4 m^{\frac{3}{2}}}{T^{\frac{3}{4}}} + \frac{|E|^{\frac{5}{2}} m^{\frac{5}{6}}}{\sqrt[4]{T}}\right) + \min_{\varepsilon \in [\frac{1}{m}, 1]} 4|E|\mathbb{E}\sqrt{2m\sum_{\mathbf{a},b \in \mathcal{C}} \log\left(\frac{\frac{1}{T}\sum_{t=1}^T b - \langle \mathbf{a}, \mathbf{c}_\varepsilon(\hat{\mathbf{x}}_t) \rangle}{\sqrt[T]{\prod_{t=1}^T b - \langle \mathbf{a}, \mathbf{c}_\varepsilon(\hat{\mathbf{x}}_t) \rangle}}\right)} + \varepsilon m \tag{93}$$

*using $k = \lceil \sqrt{T} \rceil$.*

*Proof.* Using the fact that $d = |E|$, $\nu = \mathcal{O}(|E|)$, $K = \sqrt{|E|}$, and $S_1 \leq \sum_{\mathbf{a},b \in \mathcal{C}} \frac{\|\mathbf{a}\mathbf{a}^T\|_2}{(\langle \mathbf{a}, \mathbf{1}_{|E|}/|E| \rangle - \mathbf{b})^2} = \mathcal{O}(|E|^3)$, we apply Theorem E.1 and simplify to obtain

$$\mathbb{E}\min_{\varepsilon \in [\frac{1}{m}, 1], \eta > 0} \frac{\hat{V}_\varepsilon^2}{\eta} + 32\eta |E|^2 m + \varepsilon m$$

$$+ \tilde{\mathcal{O}}\left(\frac{|E|^6 m^2}{\eta T} + \frac{|E|}{\eta k} + \frac{m}{k} + m\min\left\{\frac{\rho^2 |E|^{\frac{7}{2}}}{\eta}, \rho |E|^{\frac{11}{4}}\right\} + \frac{|E|^{\frac{11}{4}} m}{\rho}\left(\frac{1}{\sqrt{T}} + \frac{1}{\rho T}\right)\right) \tag{94}$$

Then substitute $\eta = \frac{\hat{V}_\varepsilon}{4\sqrt{2dm}} + \frac{|E|^2 \sqrt{m}}{\sqrt[4]{T}}$, set $\rho = \sqrt[4]{\frac{|E|}{T}} \sqrt[6]{m}$, and note that

$$\hat{V}_\varepsilon^2 = \sum_{\mathbf{a},b \in \mathcal{C}} \log\left(\frac{b - \langle \mathbf{a}, \mathbf{c}_\varepsilon(\hat{\hat{\mathbf{x}}}) \rangle}{\sqrt[T]{\prod_{t=1}^T b - \langle \mathbf{a}, \mathbf{c}_\varepsilon(\hat{\mathbf{x}}_t) \rangle}}\right) = \sum_{\mathbf{a},b \in \mathcal{C}} \log\left(\frac{\frac{1}{T}\sum_{t=1}^T b - \langle \mathbf{a}, \mathbf{c}_\varepsilon(\hat{\mathbf{x}}_t) \rangle}{\sqrt[T]{\prod_{t=1}^T b - \langle \mathbf{a}, \mathbf{c}_\varepsilon(\hat{\mathbf{x}}_t) \rangle}}\right) \tag{95}$$

$\square$

