# Meta-Learning Adversarial Bandit Algorithms

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

Motivated by the bandit shortest-path problem [49, 30] and described in full in Section D.3, our last application specializes Theorem D.1 to polytopes. There, the induced task-similarity is a sum across polytope boundaries, with each summand the logarithm of a quotient of arithmetic and geometric means aggregating how close the task optima are to the boundary being considered. When all distances across tasks are the same, the two means are the same and so the log of their quotient is zero, making that summand zero. Thus, the task-averaged regret improves if the optima for different tasks are at similar distances from different boundaries of the polytope.

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

-task case consider a sequence of $t = 1, \ldots, T$ shortest path instances, each with $m$ adversarial edge loss vectors $\ell_{t,i}$. The goal is to minimize average regret across instances. This setup may be viewed as learning a prediction of the optimal path, as in the algorithms with predictions paradigm in beyond-worst-case-analysis [39]; in particular, we have incorporated predictions into the algorithm of Abernethy et al. [2] via the meta-initialization approach and now present the learning-theoretic result for an end-to-end guarantee [33].

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

$\square$