# OpenReview forum: "Meta-Learning Adversarial Bandit Algorithms"
_NeurIPS.cc/2023/Conference — NeurIPS 2023 poster_

### Official Review · Reviewer_fExn · 2023-07-02

**Soundness:** 3 good
**Presentation:** 3 good
**Contribution:** 3 good
**Rating:** 6
**Confidence:** 4

**Summary:**

This paper considers learning several adversarial tasks with different loss functions simultaneously, hoping to attain better task-averaged regret if the tasks are "similar" enough (e.g., the optimal actions of all tasks concentrate on a small subset). A general meta-learning framework is derived by deploying three different algorithms as meta-learners to optimize the hyper-parameters of the base learners (OMD). It then applies to MABs and BLOs, providing various high-probability or expected regret bounds.

**Strengths:**

1. The studied problem is well-motivated. According to the authors, while the stochastic variant of this meta-learning problem is well-studied, the adversarial version tackled by this paper has never been solved.
2. The algorithmic idea is well-illustrated and thus the framework is easy to understand.
3. The framework is applied to various base-learners to prove its effectiveness, including MABs with implicit exploration or guaranteed exploration and BLOs with self-concordant barriers.

**Weaknesses:**

1. The design of the meta-learner does not look pretty exciting. Given the expression of $U_t$ in Eq. (3), it appears unsurprising that the adopted optimizers (FTL, EWOO, and MW) can be applied to their respective objectives.
In other words, it seems that the technical contribution of this paper is not solid enough -- the analyses of meta-learners and base-learners are both not novel. However, I do feel the overall result is interesting.
2. I am unsure whether the task similarity measure can generalize over the "support" of the optimizers.
Consider a special case where most of the tasks share the same optimizer but there are a few (say $\sqrt T$) outliers, would the bound scale with $\sqrt[4]{T}$ or $\mathcal O(1)$? In this case, if a meta-learner successfully rules out those outliers, the bound should scale by $\mathcal O(1)$; but using the sparsity $s$ would only give $\sqrt[4]{T}$.
Intuitively, as $H_\beta$ is taken to the average of all $\hat x$'s, such a distribution will be very different from a uniform distribution (i.e., there are $\sqrt T$ optimizers each associated with $\sqrt T$ tasks); though I'm not sure whether the current framework can capture this.
3. (minor) The bounds in the main text contain so many terms that it is hard to interpolate. Informal expressions (like Eq. (13) or Eq. (19)) can be shortened using $o_T(1)$ to highlight the main terms.

**Questions:**

See Weakness 1 & 2.

**Limitations:**

Clearly stated

---

> ### Author Rebuttal · Authors · 2023-08-10
>
> Thank you for your positive review; we hope to address your questions below.
>
> 1. [*[...] Given the expression of $U_t$ in Eq. (3), it appears unsurprising that the adopted optimizers (FTL, EWOO, and MW) can be applied to their respective objectives. [...] the analyses of meta-learners and base-learners are both not novel. However, I do feel the overall result is interesting.*]
> - While the meta-learning component of our analysis indeed builds upon existing work in the full information setting, we view the applicability of FTL and EWOO as “unsurprising” only in light of this previous work (as $U_t$ can be non-convex in $\mathbf{x}$ and non-Lipschitz in $\eta$). Furthermore, the applicability of MW does not follow from past work since it is not obvious that the first term on the RHS of (3) is Lipschitz in $\theta$; in fact, proving it requires setting-specific analysis (c.f. Lemmas B.1 & D.2). Finally, it does require substantial technical effort (c.f. Section C) in order to prove that these meta-algorithms can adapt to the similarity of true (rather than empirical) task optima, which past work does not consider because it has full information.
>
> 2. [*I am unsure whether the task similarity measure can generalize over the "support" of the optimizers. Consider a special case where most of the tasks share the same optimizer but there are a few (say $\sqrt T$) outliers, would the bound scale with $\sqrt[4]T$ or $\mathcal O(1)$? In this case, if a meta-learner successfully rules out those outliers, the bound should scale by $\mathcal O(1)$; but using the sparsity $s$ would only give $\sqrt[4]T$.*]
> - Thank you for the interesting question. While you are correct that just substituting $1+\sqrt[4]T$ for $s$ yields an undesirable bound, we *can* use our framework to obtain a result that is better than the single-task baseline of $\mathcal O(\sqrt{dm})$. In particular, in the setting you describe $H_\beta$ can be bounded by $\tilde{\mathcal O}(1+d^{1-\beta}T^{-\beta/2})$ (c.f. the bottom of this response), so for sufficiently large $T$ the bound will be $\tilde{\mathcal O}(\sqrt m)+o_T(poly(m,d))$, as desired. The caveat here is that the rate in $T$ will be quite slow. We think this is a useful analysis of our result, highlighting the advantage of not simply assuming a small set of optimal arms, and will include it (or a more general version of it, e.g. for $T^p$ outliers, $p\in[0,1]$) in revision.
>
> 3. [*[...] Informal expressions (like Eq. (13) or Eq. (19)) can be shortened using $o_T(1)$ to highlight the main terms.*]
> - In general we agree, although note that keeping the rates in $T$ can be useful for comparison with other works, e.g. Azizi et al. [10].
>
> ### Bound on $H_\beta$ in the presence of $\sqrt T$ outliers
>
> Suppose all but $\sqrt T$ tasks have optimal action $a^\ast\in[d]$.
> Then applying Claim A.1 and the mean-as-minimizer property of Bregman divergences we have that
> $$\begin{align}
> H_\beta(\hat{\bar x})
> &=-\psi_\beta(\hat{\bar x}) \\\\
> &=\frac1T\sum_{t=1}^T\psi_\beta(\hat x_t)-\psi_\beta(\hat{\bar x}) \\\\
> &=\frac1T\sum_{t=1}^TB_\beta(\hat x_t||\hat{\bar x}) \\\\
> &=\min_{x\in\triangle_d}\frac1T\sum_{t=1}^TB_\beta(\hat x_t||x) \\\\
> &\le\min_{\delta\in(0,1)}\frac1T\sum_{t=1}^TB_\beta(\hat x_t||(1-\delta)e_{a^\ast}+\delta 1_d/d) \\\\
> &=\min_{\delta\in(0,1)}\frac1T\sum_{t=1}^T\frac1{1-\beta}\sum_{a=1}^d((1-\delta)1_{a=a^\ast}+\delta/d)^\beta-\hat x_{t[a]}^\beta+\frac{\beta(\hat x_{t[a]}-(1-\delta)1_{a=a^\ast}-\delta/d)}{((1-\delta)1_{a=a^\ast}+\delta/d)^{1-\beta}} \\\\
> &=\min_{\delta\in(0,1)}\frac1T\sum_{t=1}^T\sum_{a=1}^d((1-\delta)1_{a=a^\ast}+\delta/d)^\beta-\frac{\hat x_{t[a]}^\beta}{1-\beta}+\frac{\beta\hat x_{t[a]}^\beta}{(1-\beta)((1-\delta)1_{a=a^\ast}+\delta/d)^{1-\beta}} \\\\
> &\le\min_{\delta\in(0,1)}\delta^\beta d^{1-\beta}+\frac\beta{(1-\beta)T}\sum_{t=1}^T\sum_{a=1}^d\frac{\hat x_{t[a]}^\beta}{((1-\delta)1_{a=a^\ast}+\delta/d)^{1-\beta}}
> \end{align}$$
> For non-outlier tasks we have $a=a^\ast$ so the last summation over $a$ is at most $1/(1-\delta)^{1-\beta}$, while for outlier tasks it is at most $(d/\delta)^{1-\beta}$.
> Since we have $T-\sqrt T\le T$ of the former and $\sqrt T$ of the latter we have the bound
> $$H_\beta\le\min_{\delta\in(0,1)}\delta^\beta d^{1-\beta}+\frac\beta{(1-\beta)(1-\delta)^{1-\beta}}+\frac{\beta(d/\delta)^{1-\beta}}{(1-\beta)\sqrt T}$$
> Assuming $\beta\in[\frac1{\log d},\frac12]$ this means $H_\beta=\tilde{\mathcal O}(1+d^{1-\beta}T^{-\beta/2})$.

---

> > ### Comment · Reviewer_fExn · 2023-08-18
> >
> > Thanks for your clarification. I decide to keep my recommendation unchanged.

---

### Official Review · Reviewer_WoUK · 2023-07-03

**Soundness:** 3 good
**Presentation:** 3 good
**Contribution:** 3 good
**Rating:** 7
**Confidence:** 2

**Summary:**

This paper focused on online meta-learning with bandit feedback, and developed and applied a meta-algorithm for learning to initialize and tune bandit algorithms, obtaining task-average regret guarantees for both MAB and linear bandits.  Specifically, a meta-algorithm was developed for learning the variants of OMD, which was further applied to OMD with the Tsallis regularizer. Furthermore, the meta-algorithm was adapted to the adversarial BLO problem via setting the regularizer to be a self-concordant barrier function.

**Strengths:**

- This paper is the first to consider meta-learning under adversarial bandit feedback.
- A meta-algorithm was designed for learning the variants of OMD, which can simultaneously tune the initialization and other hyperparameters.
- Strong theoretical performance guarantees were presented for the proposed algorithm.
- The paper is well-written and easy to follow though it is heavy in theory.

**Weaknesses:**

- This paper mainly focused on the pure adversarial settings.  As the authors mentioned, there are extensive works studied the stochastic settings, and given the popularity of "best-of-both-worlds" settings in the community, can you envision what is the major challenges to generalize the solutions to the best-of-both-worlds settings, from both the algorithmic and performance analysis perspectives?
- As the paper considered the applications of the proposed algorithms, it may be interesting to do some case studies with real-world applications, rather than only on theoretical applications.

**Questions:**

- This paper mainly focused on the pure adversarial settings.  As the authors mentioned, there are extensive works studied the stochastic settings, and given the popularity of "best-of-both-worlds" settings in the community, can you envision what is the major challenges to generalize the solutions to the best-of-both-worlds settings, from both the algorithmic and performance analysis perspectives?

**Limitations:**

n/a societal impact.

---

> ### Author Rebuttal · Authors · 2023-08-10
>
> Thank you for your positive review; we hope to address your questions and concerns below.
>
> 1. [*[C]an you envision what is the major challenges to generalize the solutions to the best-of-both-worlds settings, from both the algorithmic and performance analysis perspectives?*]
> - One challenge is that the online-within-online setting has two parts, each of which can be either stochastic or adversarial, giving rise to four settings that need to be supported. In order to prove that the solution is “best” for each of these settings, four lower bounds will need to be proved first. Additionally, dependence on the true optima-in-hindsight (Section 3.2) will require a best-arm identification procedure that is best-of-both-worlds and simultaneously works alongside the regret-minimization algorithm.
>
>
> 2. [*As the paper considered the applications of the proposed algorithms, it may be interesting to do some case studies with real-world applications, rather than only on theoretical applications.*]
> - We agree that it will be interesting to develop novel real-world algorithms that are inspired by this theoretical framework. However, our goal in this work was theoretical depth and breadth, with the application of our framework to four different bandit algorithms (Sections 3.1, 3.2, 4, and D.3) across three different geometries (simplex, sphere, and polytope). Thus we view real-world applications as out of scope for us due to constraints on space and the need for conciseness, and leave it for future work.

---

> > ### Comment · Reviewer_WoUK · 2023-08-17
> >
> > Thank you for the clarification.

---

### Official Review · Reviewer_zCMR · 2023-07-04

**Soundness:** 3 good
**Presentation:** 3 good
**Contribution:** 3 good
**Rating:** 6
**Confidence:** 3

**Summary:**

In this paper, the authors consider an online meta-learning problem with the adversarial online-with-online partial information setting,
where a learner selects parameters across T tasks with m rounds and can utilize similarity between tasks to achieve a low regret.
First, the authors propose a meta algorithm which optimizes hyperparameters of the inner loop algorithm (OMD).
As an instance of this algorithm, the authors consider OMD for MAB and BLO (bandit linear optimization).
Then, they prove several upper bounds of the average regret across $T$ tasks using a similarity of optimal parameters across tasks.

**Strengths:**

1. While existing papers considered online-with-online meta-learning in stochastic or full-information setting, this paper considers the adversarial bandit feedback setting, which is practically important.
2. Although the proposed algorithm is based on one in the full information setting, there is novelty due to the bandit feed back (e.g., estimation of true $H_\beta$).
3. The authors provide regret upper bounds under several settings and results seems valid.

**Weaknesses:**

1. There is no discussion on lower bounds. Therefore, it is not clear whether the proposed method is optimal.

**Questions:**

1. In the related work section, the authors compare their algorithm to algorithm for dynamic regret optimization, but they discuss only the number of switches. There are algorithms whose regret bounds involve total variance of the environment (e.g. $\Delta$ in Wei, Chen-Yu, and Haipeng Luo. "Non-stationary reinforcement learning without prior knowledge: An optimal black-box approach." Conference on learning theory, 2021.). Are these algorithms related to the problem setting in Sec 3.2? Is it possible to provide a comparison?
2. In the stochastic setting, is there a known lower bound?

**Limitations:**

Yes

---

> ### Author Rebuttal · Authors · 2023-08-10
>
> Thank you for your positive review; we hope to address your questions and concerns below.
>
> Weaknesses:
> 1. [*There is no discussion on lower bounds. Therefore, it is not clear whether the proposed method is optimal.*]
> - While we agree that lower bounds can be informative, our goal in this paper was to demonstrate the potential benefits of meta-learning across a broad number of settings, including four different bandit algorithms (Sections 3.1, 3.2, 4, and D.3) and three different geometries (simplex, sphere, and polytope). See also our response to your question on stochastic lower bounds below.
>
>
>
> Questions:
> 1. [*In the related work section, the authors compare their algorithm to algorithm for dynamic regret optimization, but they discuss only the number of switches. There are algorithms whose regret bounds involve total variance of the environment (e.g. $\Delta$ in [Wei & Luo (COLT 2021)].). Are these algorithms related to the problem setting in Sec 3.2? Is it possible to provide a comparison?*]
> - Thank you for the reference, which we will add to our related work. Note that Wei & Luo (2021) study several different stochastic settings (from MAB to MDP), whereas we focus on the adversarial setting; thus it is not obvious how to generalize their $\Delta$ to our setting, as it quantifies changes in the loss distributions over time.  From looking at the dynamic regret bound they highlight in the abstract—$\tilde{\mathcal O}(\min\{\sqrt{LT},\Delta^\frac13T^\frac23\})$, where $L$ is equal to the number of changes in distribution ($T$ in our setting), $T$ corresponds to the number of rounds ($mT$ in our setting), and $\Delta$ is a type of distributional path length—it seems that their algorithms will have task-averaged regret $\tilde{\mathcal O}(\min\{\sqrt m,\Delta^\frac13m^\frac23/T^\frac13\})$. This will do well if the average change in distributions across tasks is sublinear in $T$ (i.e. $(\Delta/T)^\frac13=o_T(1)$); on the other hand, if the environment switches between a small set of different tasks (the sparse setting where our MAB algorithm does well) then $(\Delta/T)^\frac13=\Theta(1)$ and their approach will not improve upon the baseline of $\tilde{\mathcal O}(\sqrt m)$. Even in this case, of course, their algorithm has the advantage of not needing to know the location of the task switches.
>
> 2 [*In the stochastic setting, is there a known lower bound?*]
> - There are known lower bounds for some specific multi-task stochastic bandit settings [1,2,4,5], but to our knowledge the only one in the sequential setting across tasks is in the full information adversarial setting [3].
>
> [1] Azizi, Kveton, Ghavamzadeh, Katariya. *Meta-learning for simple regret minimization.* AAAI 2023.
> [2] Cella, Pontil. *Multi-task and meta-learning with sparse linear bandits.* UAI 2021.
> [3] Khodak, Balcan, Talwalkar. *Provable guarantees for gradient-based meta-learning.* ICML 2019.
> [4] Simchowitz, Tosh, Krishnamurthy, Hsu, Lykouris, Dudik, Schapire. *Bayesian decision-making under misspecified priors with applications to meta-learning.* NeurIPS 2021.
> [5] Yang, Hu, Lee, Du. *Impact of representation learning in linear bandits.* ICLR 2021.

---

> > ### Comment · Reviewer_zCMR · 2023-08-18
> >
> > Thank you for clarifications. I would like to keep the current scores.

---

### Official Review · Reviewer_qSEU · 2023-07-04

**Soundness:** 3 good
**Presentation:** 3 good
**Contribution:** 3 good
**Rating:** 6
**Confidence:** 2

**Summary:**

The paper proposes an online mirror descent approach for adversarial bandit feedback-based online meta learning, utilizing FTL for initialization, EWOO for step-size, and MW for regularizer-specific parameters. The proposed method is applied to two widely adopted applications: multi-armed bandits (MAB) with Tsallis regularizer and bandit linear optimization (BLO) with self-concordant barrier function regularizer. The authors provide theoretical analysis on asymptotic task-averaged regret for MAB and BLO.



**Strengths:**

The paper is well-written and seems to cover an important topic in the community. I check some proofs, and they seems correct.


**Weaknesses:**

While the paper is theoretical in nature, it would be beneficial to include some empirical results to further validate the proposed approach.



**Questions:**

The reviewer is not very familiar with this topic and would like to see the comments and feedback from other reviewers before asking further questions.

**Limitations:**

It would be nice to provide empirical results ( it is not mandatory and the paper's contribution is still valuable without them).

---

> ### Author Rebuttal · Authors · 2023-08-10
>
> Thank you for your positive review; we are happy to answer additional questions later. With respect to your concern about experiments: our goal was theoretical depth and breadth, with the application of our framework to four different bandit algorithms (Sections 3.1, 3.2, 4, and D.3) across three different geometries (simplex, sphere, and polytope). Thus we view experimental contributions as out of scope for us due to constraints on space and the need for conciseness.

---

> > ### Comment · Reviewer_qSEU · 2023-08-17
> >
> > I have read the authors' response and other reviewers' comments. I am glad to keep my score.

---

### Official Review · Reviewer_yLU7 · 2023-07-24

**Soundness:** 3 good
**Presentation:** 3 good
**Contribution:** 3 good
**Rating:** 6
**Confidence:** 2

**Summary:**

The paper discusses online meta-learning with bandit feedback to enhance performance across multiple tasks that share a natural similarity measure. The study focuses on the adversarial online-within-online partial-information setting and proposes meta-algorithms that combine outer learners to optimize the initialization and other hyperparameters of an inner learner for two cases: multi-armed bandits (MAB) and bandit linear optimization (BLO). For MAB, the meta-learners use the Tsallis-entropy generalization of Exp3, and the task-averaged regret improves when the entropy of the optima-in-hindsight is small. For BLO, the approach involves learning to initialize and tune online mirror descent (OMD) with self-concordant barrier regularizers, where the task-averaged regret is related to an action space-dependent measure induced by these regularizers. The guarantees provided in the study are based on demonstrating that unregularized follow-the-leader combined with two levels of low-dimensional hyperparameter tuning can effectively learn a sequence of affine functions of non-Lipschitz and sometimes non-convex Bregman divergences, which bound the regret of OMD.

**Strengths:**

This paper designs a meta-algorithm combining FTL, EWOO, MW to set the initialization, step-size, and the regularizer. Then, it has to direct applications for MAB and BLO. Then, the authors provide rigorous average regret analysis with Tsallis entropy, which are the first results for online learning with Bregman divergences.

**Weaknesses:**

It will be better if authors can provide some real-world applications that the designed algorithm can adapt to.

**Questions:**

(1) For the MAB algorithms, there may be only one served task in each round, i.e., observe the loss. In this case, how to run algorithm 1 and do we need to update the parameters of all tasks?

**Limitations:**

.

---

> ### Author Rebuttal · Authors · 2023-08-10
>
> Thank you for your positive review; we hope to address your questions and concerns below.
>
> 1. [*It will be better if authors can provide some real-world applications that the designed algorithm can adapt to.*]
> -  As noted in the introduction, single-task bandits are widely used in applications such as recommender systems and experimental design; multi-task versions of these applications arise naturally. For example, a recommendation engine may face a sequence of different users with multiple interactions each, or a scientist may run a sequence of similar multi-round experiments.
>
> 2. [*For the MAB algorithms, there may be only one served task in each round, i.e., observe the loss. In this case, how to run algorithm 1 and do we need to update the parameters of all tasks?*]
> - If we are interpreting your question correctly, it concerns the somewhat different setting of *multi-task* online learning (Dekel, Long, & Singer, COLT 2006), where on each round we see a loss associated with one of T tasks, rather than *meta* online learning, where we see T tasks one-after-another. The multi-task setting has generally required rather different algorithms.

---

> > ### Comment · Reviewer_yLU7 · 2023-08-17
> >
> > Thanks for the response. I keep my original assessment.

---

### Official Review · Reviewer_eUxe · 2023-07-26

**Soundness:** 3 good
**Presentation:** 2 fair
**Contribution:** 3 good
**Rating:** 6
**Confidence:** 2

**Summary:**

This paper introduces a meta-learning algorithmic framework for adversarial bandits, designed to fine-tune the initialization and hyperparameters of the internal learner, thereby enhancing performance across multiple tasks. The efficacy of this approach is validated via a theoretical analysis on multi-armed bandits and linear bandits. The authors also propose a task-similarity measure with significant implications concerning entropy and proximity to the task boundary.

**Strengths:**

- The paper addresses meta-learning under adversarial bandit feedback, which is an interesting problem by itself, and the meta-learning approach poses a solid contribution to the bandit community, with a new notion of task-similarity based on entropy.
- The concrete steps towards building the algorithms is clear and statements in the paper seems solid.


**Weaknesses:**

- The paper does not have any numerical validations of the proposed algorithm. This puts doubts on the practicality of the proposed framework and methods. Synthetic as well as real-world examples should be tested upon, and comparisons against meta learning stochastic bandits should be discussed in the numerical experiments as well.
- The computational complexity is not adequately addressed within the main body of the paper. Specifically, the computational overhead associated with the Multiplicative Weights (MW) method could be prohibitively high with a large grid and action space.
- The interpretation of task-similarity lacks a sufficient explanation of its underlying motivation. Even in application cases where the authors justify the task-similarity measure via average entropy or proximity to the decision boundary, it remains unclear why this measure was chosen and how it compares to other task-similarity measures in stochastic bandit scenarios (and/or those with contamination or parameter noise). As it stands, the presentation in the paper makes it seem more like an artifact of the regret proof.

**Questions:**

- Could the authors elaborate on the connections to meta-learning in adversarial Reinforcement Learning (RL)? While meta-learning for adversarial bandits appears to be a novel concept, meta-learning in adversarial RL, a generalization of bandits, has been explored in the literature, as evidenced by [1] for example.
- Could the authors provide insights on how the algorithm's implementation could be optimized for efficiency?
- Could the authors demonstrate the effectiveness of the model and the algorithm in a well-motivated real-world setting through numerical experiments?
- Could the authors offer a more intuitive explanation of the task-similarity measure and how it compares to existing methods, beyond comments about being 'distributional assumption-free'?

[1] Lin, Zichuan, Garrett Thomas, Guangwen Yang, and Tengyu Ma. "Model-based adversarial meta-reinforcement learning." Advances in Neural Information Processing Systems 33 (2020): 10161-10173.

**Limitations:**

The authors have acknowledged the limitations of their work, including the absence of a gap-free task similarity criterion, the assumption of known task boundaries, and the lack of a 'best-of-both-worlds' guarantee.

---

> ### Author Rebuttal · Authors · 2023-08-10
>
> Thank you for your review; we address your questions and concerns below.
>
> Weaknesses:
> 1. [*The paper does not have any numerical validations of the proposed algorithm. [...]*]
> - Our goal in this work was theoretical depth and breadth, with the application of our framework to four different bandit algorithms (Sections 3.1, 3.2, 4, and D.3) across three different geometries (simplex, sphere, and polytope). This puts experimental contributions out of scope for us due to constraints on space and the need for conciseness. As noted by Reviewer qSEU, they are “not mandatory and the paper's contribution is still valuable without them.”
>
> 2. [*The computational complexity is not adequately addressed within the main body of the paper. Specifically, the computational overhead associated with the Multiplicative Weights (MW) method could be prohibitively high with a large grid and action space.*]
> - While it is in the appendix due to space constraints, our thorough examination of both computational and space complexity in Section A.3 does discuss the overhead of MW; per-iteration its cost is sublinear in T and d, which we do not view as prohibitively high.
>
> 3. [*The interpretation of task-similarity lacks a sufficient explanation of its underlying motivation. [... I]t remains unclear why this measure was chosen and how it compares to other task-similarity measures in stochastic bandit scenarios [...] As it stands, the presentation in the paper makes it seem more like an artifact of the regret proof.*]
> - We view task-similarity measures that arise from distance measures or entropies associated directly with the algorithms being run on those tasks to be inherently natural, rather than simply proof artifacts. For MAB we provide an interpretation of the task-similarity measure below Corollary 3.1 and *directly* compare it to the one that Azizi et al. [10] used for the stochastic bandit setting at the end of Section 3.1. The fact that our results yield meaningful guarantees under their “small set of optimal arms” assumption, which we also highlight in the introduction, is strong evidence of the usefulness of our approach to measuring task similarity. For BLO we provide an interpretation of the task-similarity measure below Corollary 4.1.
>
> Questions:
> 1. [*Could the authors elaborate on the connections to meta-learning in adversarial Reinforcement Learning (RL)? While meta-learning for adversarial bandits appears to be a novel concept, meta-learning in adversarial RL, a generalization of bandits, has been explored in the literature, as evidenced by [1] for example.*]
> - In RL the losses are stochastic and depend on a state in an MDP, whereas in adversarial bandits the losses are chosen adversarially; thus RL does not generalize adversarial bandits. In the referenced work [1], it is the “meta” aspect that is adversarial, not the RL. Specifically, they study a batch setting across tasks and propose that the learner should choose the tasks using adversarial training in order to do well in the case of distribution shift, *but there is no actual adversary*; our work studies an online setting across tasks where an adversary chooses the tasks.
>
> 2. [*Could the authors provide insights on how the algorithm's implementation could be optimized for efficiency?*]
> - This is discussed in Section A.3; in-particular, we suggest that replacing our use of EWOO by MW may increase efficiency at the cost of worse regret. Please also see our response to your second weakness above.
>
> 3. [*Could the authors demonstrate the effectiveness of the model and the algorithm in a well-motivated real-world setting through numerical experiments?*]
> - Please see our response to your first weakness above.
>
> 4. [*Could the authors offer a more intuitive explanation of the task-similarity measure and how it compares to existing methods, beyond comments about being 'distributional assumption-free'?*]
> - Please see our response to your third weakness above. Note also that “distributional assumption-free” is an aspect of our guarantees, rather than of our task-similarity measure.

---

> > ### Comment · Reviewer_eUxe · 2023-08-13
> >
> > I thank the authors for the clarification and hence increase my score from 4 to 6.

---

### Author Rebuttal · Authors · 2023-08-10

We would like to thank the reviewers for their helpful reviews, which among other things have led to several useful references and an analysis of the robustness of our result to outliers (c.f. [our response to Reviewer fExn](https://openreview.net/forum?id=r6xGZ0XL2g&noteId=hvQFk8uNIu)); we plan to incorporate these into the revision. We hope to address any follow-up questions in the discussion.

To summarize the contributions of our paper: we develop a meta-algorithm for meta-learning the parameters of adversarial bandit methods and apply it to four such procedures (Sections 3.1, 3.2, 4, and D.3) across three different geometries (simplex, sphere, and polytope). This is the first analysis of meta-learning for adversarial bandits, and it yields provable performance improvements for sequences of multiple similar tasks, where performance is measured by task-averaged regret and task similarity is measured by distances between task optima induced by the specific within-task algorithm being studied. As an example, in the case of $T$ multi-armed bandit tasks with $m$ rounds and $d$ arms each where only $s\ll d$ of the arms are ever optimal for any task, the average regret of our approach is $O(\sqrt{sm\log d})$ (ignoring lower order terms in $m$ and $T$), compared to the usual MAB regret of $O(\sqrt{dm})$.

---

### Decision · Program_Chairs · 2023-09-21

**Decision:**

Accept (poster)

**Comment:**

This work studies meta-learning parameters of adversarial bandit algorithms and proposes a regret bound on the average regret of adversarial bandit algorithms for $T$ bandit problems. The regret bound depends on the distribution of optimal arms and is favorable in settings where the total number of optimal arms is significantly smaller than the number of arms for each task. Reviewers are overall positive on the paper and agree that the paper addresses an interesting problem, giving a novel solution with meaningful guarantees. Most reviewers also agree that the paper is well written and the presentation is good. Multiple reviewers have expressed concern about lack of empirical evaluation of the results and while I agree that this is a theory paper, if the authors are able to add experiments, even as part of the appendix, this would strengthen the results. Multiple reviewers have also noted that computational complexity is not discussed in the main paper. Authors are encouraged to have a brief discussion in the camera ready version, space permitting.